# Provably Safe Neural Network Controllers
# via Differential Dynamic Logic

**Samuel Teuber**[1]    **Stefan Mitsch** [2]    **André Platzer** [1,3]

[1] Karlsruhe Institute of Technology    [2] DePaul University    [3] Carnegie Mellon University

teuber@kit.edu   smitsch@depaul.edu   platzer@kit.edu

## Abstract

While neural networks (NNs) have a large potential as autonomous controllers for Cyber-Physical Systems, verifying the safety of *neural network based control systems (NNCSs)* poses significant challenges for the practical use of NNs—especially when safety is needed for *unbounded time horizons*. One reason for this is the intractability of analyzing NNs, ODEs and hybrid systems. To this end, we introduce VerSAILLE (**Ver**ifiably **S**afe **AI** via **L**ogically **L**inked **E**nvelopes): The first general approach that allows reusing control theory literature for NNCS verification. By joining forces, we can exploit the efficiency of NN verification tools while retaining the rigor of differential dynamic logic (dL). Based on a provably safe control envelope in dL, we derive a specification for the NN which is proven with NN verification tools. We show that a proof of the NN's adherence to the specification is then *mirrored* by a dL proof on the infinite-time safety of the NNCS.

The NN verification properties resulting from hybrid systems typically contain *nonlinear arithmetic* over formulas with *arbitrary logical structure* while efficient NN verification tools merely support linear constraints. To overcome this divide, we present Mosaic: An efficient, *sound and complete* verification approach for polynomial real arithmetic properties on piece-wise linear NNs. Mosaic partitions complex NN verification queries into simple queries and lifts off-the-shelf linear constraint tools to the nonlinear setting in a completeness-preserving manner by combining approximation with exact reasoning for counterexample regions. In our evaluation we demonstrate the versatility of VerSAILLE and Mosaic: We prove infinite-time safety on the classical Vertical Airborne Collision Avoidance NNCS verification benchmark for some scenarios while (exhaustively) enumerating counterexample regions in unsafe scenarios. We also show that our approach significantly outperforms the State-of-the-Art tools in closed-loop NNV.

## 1   Introduction

For controllers of Cyber-Physical Systems (CPSs), the use of neural networks (NNs) is both a blessing and a curse. On the one hand, using NNs allows the development of goal-oriented controllers that optimize soft requirements such as passenger comfort, frequency of collision warnings or energy efficiency. On the other hand, guaranteeing that *all* control decisions chosen by an NN are safe is very difficult due to the complex feedback loop between the subsymbolic reasoning of an NN and the intricate dynamics often encountered in physical systems. How can this curse be alleviated? Neural Network Verification (NNV) techniques all have tried one of three strategies: Open-loop NNV entirely omits the analysis of the physical system and only analyzes input-output properties of the NN [12, 23, 43, 44, 58, 59, 95, 98–100]. Open-loop analyses alone cannot justify the safety of an NNCS, because they ignore its physical, feedback-loop dynamics. Closed-loop NNV performs a time-bounded analysis of the feedback loop between the NN and its physical environment [5, 18, 34, 37, 47–49, 85, 88, 93, 94, 97]. Unfortunately, a safety guarantee that comes with a time-bound (measured in seconds

38th Conference on Neural Information Processing Systems (NeurIPS 2024).

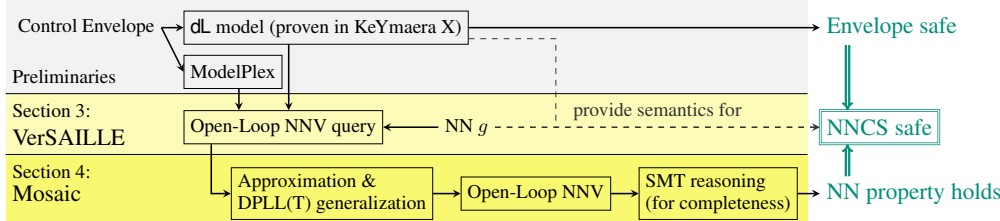

Figure 1: VerSAILLE reflects a proof of a control envelope in an NN to verify infinite-time safety of an NNCS from mere open-loop NNV properties. Mosaic *completely* lifts off-the-shelf open-loop NNV tools to polynomial arithmetic by combining approximation with judicious SMT reasoning.

rather than minutes or hours) is often insufficient when it comes to deploying safety-critical NNCSs in the real world. For example, the safety of an adaptive cruise control system must be independent of the trip length. Finally, another line of work explored techniques for learning and then verifying *approximations of* barrier certificates for infinite-time guarantees [3,9,25,26,31,64]. For *continuous-time*, verification has not been scaled beyond simple linear control functions [25, Appendix] as it requires open-loop NNV w.r.t. nonlinear specifications, which is a notoriously neglected topic [30,31].

As an alternative to the three outlined approaches, we propose to verify NNCSs based on the rigorous mathematical foundations of differential dynamic logic (dL). dL is a program logic allowing the proof of infinite-time safety for abstract, nondeterministic control strategies (often called control envelopes). Due to its expressiveness and its powerful proof calculus, dL even allows the derivation of such guarantees for continuous-time systems or systems whose differential equations have no closed-form solution. By grounding our verification approach in dL, we can reuse safety results from the control theory literature for NN verification – especially for cases where characterizations of safe behavior and controllable/invariant regions are known (e.g. airborne collision avoidance [56]). How this knowledge can be reused is a non-trivial question: While dL is an excellent basis for reasoning about symbolic control strategies, the numerical/subsymbolic reasoning of NNs at their scale is far beyond the intended purpose of dL's proof calculus. Conversely, open/closed-loop NNV tools and barrier certificates lack the infinite-time and exact reasoning available within dL. This work demonstrates how open-loop NNV can be combined with dL reasoning to combine their strengths while canceling out their weaknesses. Consequently, by relying on results from the control theory literature, we prove infinite-time safety guarantees for NNCSs that are not provable through either technique alone.

**Overview.** This paper alleviates the curse of NNCS safety. As shown in Figure 1, our work integrates the deductive approach of dL with techniques for open-loop NNV. To apply our approach, we assume that an abstract, nondeterministic control envelope has already been verified in dL (via KeYmaera X [38], synthesized via CESAR [57] or from the literature). Based on the dL safety result, VerSAILLE (**Ver**ifiably **S**afe **AI** via **L**ogically **L**inked **E**nvelopes; Section 3) derives a verification query for open-loop NNV by instrumenting ModelPlex [69]. By reflecting the NN through a *mirror program* in dL, we can then reason about an NNCS in- and outside the dL calculus simultaneously. The verification of an open-loop NNV query generated by VerSAILLE yields a dL proof that the NNCS refines a safe control envelope —implying that the infinite-time safety guarantee carries over to the NNCS.

Due to the inherent nonlinearities of hybrid systems, the generated open-loop NNV queries often contain polynomial arithmetic and the formulas have arbitrary logical structure. Hence for such queries, we also introduce Mosaic—an efficient, *sound and complete* framework for open-loop NNV tools. The approach *lifts* complete off-the-shelf open-loop NNV tools for linear constraints to *polynomial* constraints with *arbitrary* logical structure. To this end, we combine approximation with a generalization of DPLL(T) that makes the logical decomposition *efficiently* applicable to NN verification (whereas "classical" DPLL(T) would become prohibitively inefficient). At the same time, Mosaic retains completeness by generalizing counterexamples into locally affine regions (Section 4). In summary, VerSAILLE provides rigorous semantics and a formal proof of infinite-time safety, while Mosaic makes our approach practically applicable to real-world systems (see also Section 5).

**Contribution.** Our contribution has three parts. While our implementation ($N^3V$) supports NNs most commonly analyzed by open-loop NNV ($\mathrm{ReLU}$ NNs), our theoretical contribution (VerSAILLE) reaches far beyond this and lays the foundations for analyzing a wide range of NNCS architectures:

- We present VerSAILLE, the formal foundation that, for the first time, enables a *sound proof of infinite-time safety for a concrete NNCS* by reusing safety proofs from control-theory literature (in the form of dL models). VerSAILLE supports a large class of feed-forward NNs (any NN with piece-wise Noetherian activation functions; see Section 2).

- We introduce Mosaic, a framework for the efficient, *sound and complete* verification of properties in *polynomial* real arithmetic on piece-wise linear NNs. Unlike other NN verifiers, Mosaic furthermore supports constraints with *arbitrary* propositional structure. Mosaic combines approximation techniques, a generalization of DPLL(T), and judicious SMT reasoning to *lift* sound and complete linear-constraint open-loop NNV tools to efficient, *sound and complete* polynomial constraint verification. Mosaic can exhaustively characterize unsafe state space regions (useful for retraining or the generation of fallback controllers).

- We implement Mosaic for $\mathrm{ReLU}$ NNs in the tool $N^3V$ and demonstrate our approach on three case studies from adaptive cruise control, airborne collision avoidance ACAS X and steering under uncertainty. We show that, unlike $N^3V$, State-of-the-Art closed-loop NNV tools cannot provide infinite-time guarantees due to approximation errors.

**Running Example.** The common NNCS safety benchmark of Adaptive Cruise Control [22, 39, 51] will serve as running example to demonstrate the introduced concepts. Consider an ego-car following a front-car on a 1-D lane as shown in Figure 2. The front-car drives with constant velocity $v_{\mathrm{const}}$ while the ego-car (at position $p_{\mathrm{rel}}$ behind the front-car) approaches with arbitrary initial (relative) velocity $v_{\mathrm{rel}}$ which is adjusted through the ego-car's acceleration $a_{\mathrm{rel}}$. The primary objective is to ensure the ego-car never crashes into the front car (i.e. $p_{\mathrm{rel}} > 0$), however there may be secondary objectives

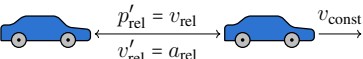

Figure 2: Adaptive Cruise Control: The front-car (right) drives with constant $v_{\mathrm{const}}$. The ego-car approaches with relative velocity $v_{\mathrm{rel}}$ (controlled via $a_{\mathrm{rel}}$) from $p_{\mathrm{rel}}$.

(e.g. energy efficiency) which are learned by an NNCS. We demonstrate how a nondeterministic, high-level acceleration strategy (i.e. a safe envelope) can be modeled and verified in dL (Section 2), how VerSAILLE derives NN properties (Section 3) and how such polynomial properties can be verified on a given NN (Section 4). No techniques are specific to the running example, but all are applicable to a wide range of NNCSs—as demonstrated by our evaluation (Section 5).

## 2 Background

We review dL, NNs and NN verification. $\mathrm{FOL}_{\mathbb{R}}$ (resp. $\mathrm{FOL}_{\mathrm{L}\mathbb{R}}$) is the set of polynomial (resp. linear) real arithmetic first-order logic formulas. $\mathrm{FOL}_{\mathrm{N}\mathbb{R}}$ extends $\mathrm{FOL}_{\mathbb{R}}$ with *Noetherian functions* [78] $h_1, \ldots, h_r$. A *Noetherian chain* is a sequence of real analytic functions $h_1, \ldots, h_q$ s.t. all partial derivatives of all $h_j$ can be written as a polynomial $\frac{\partial h_j(y)}{\partial y_i}(y) = p_{ij}(y, h_1(y), \ldots, h_q(y))$. *Noetherian functions* are representable as a polynomial over functions in a Noetherian chain. Most activation functions used in NNs are Noetherian (see Appendix A.2) Atoms of a formula $\zeta$ are denoted as $\mathrm{Atom}(\zeta)$ and its variables as $\mathrm{V}(\zeta)$.

| Program | Semantics |
|---|---|
| $x := e$ | Assign term $e$ to $x$ |
| $x := *$ | Nondet. assign to $x$ |
| $?Q$ | Test of formula $Q$ |
| $x' = t \& Q$ | Evolve $x$ along the diff. equation within $Q$ |
| $\alpha \cup \beta$ | Nondet. choice |
| $\alpha; \beta$ | Sequential composition |
| $(\alpha)^*$ | Nondet. loop |

Table 1: Program primitives of dL

### 2.1 Differential Dynamic Logic

Differential dynamic logic (dL) [74–76, 78] is a first-order multi-modal logic in which the modality is parameterized with a *hybrid program* describing a (discrete or continuous) state transition (see also Appendix A). Thus, dL formulas are evaluated in a state $\nu$ (if $\nu$ satisfies a formula $\psi$ we denote this as $\nu \vDash \psi$). Hybrid Programs are constructed from the primitives in Table 1 and are first-class citizens of the logic (see example below). dL is tailored to the analysis of (time discrete and time continuous) hybrid systems and supports the analysis of differential equations. Through its invariance reasoning capabilities, dL allows us to prove the infinite-time safety of control envelopes w.r.t. a system's dynamics—even for cases where the dynamics' differential equations have no closed-form solution. There is a large body of research on the verification of real-world control envelopes using dL (e.g.

ACAS X [50]). In dL, the formula $[\alpha]\psi$ expresses that $\psi$ is always satisfied after the execution of $\alpha$ and $\langle\alpha\rangle\psi$ that there exists a state satisfying $\psi$ after the execution of $\alpha$. dL comes with a sound and relatively complete proof calculus [74, 76, 78] and an interactive theorem prover KeYmaera X [38].

**Running Example.** We model our running example as a hybrid program in dL with differential equations describing the evolution of $p_{\text{rel}}, v_{\text{rel}}, a_{\text{rel}}$ along with a *control envelope*, i.e. an abstract acceleration strategy, $\alpha_{\text{ctrl}}$ that runs at least every $T$ seconds while the overall system may run for arbitrarily many iterations (modeled by a nondeterministic loop). Given suitable initial conditions (accInit), our objective is to prove the absence of collisions ($p_{\text{rel}} > 0$). This can be achieved by proving Formula (1) where the place-holder $\alpha_{\text{ctrl}}$ determines the relative acceleration $-B \leq a_{\text{rel}} \leq A$ ($A$ and $-B$ are resp. maximal acceleration/braking).

$$\underbrace{\text{accInit}}_{\text{initial conditions}} \rightarrow \Big[ \big( \underbrace{\alpha_{\text{ctrl}}}_{\text{controller}} ; c \coloneqq 0; \underbrace{(p'_{\text{rel}} = v_{\text{rel}}, v'_{\text{rel}} = -a_{\text{rel}}, c' = 1 \,\&\, c \leq T)}_{\text{environment}} \big)^* \Big] \underbrace{p_{\text{rel}} > 0}_{\text{safety constraint}} \quad (1)$$

Our control envelope $\alpha_{\text{ctrl}}$ allows braking with $-B$ and an acceleration of 0 or another value if the constraint accCtrl$_0$ or resp. accCtrl$_1$ is satisfied (the concrete constraints are found in Appendix D):

$$\alpha_{\text{ctrl}} \equiv a_{\text{rel}} \coloneqq -B \cup (a_{\text{rel}} \coloneqq 0; ?\,(\text{accCtrl}_0)) \cup (a_{\text{rel}} \coloneqq *; ?\,(-B \leq a_{\text{rel}} \leq A \wedge \text{accCtrl}_1))$$

The envelope is nondeterministic: While always braking with $-B$ would be safe, an NNCS may *learn* to balance braking with secondary objectives (e.g. minimal acceleration or not falling behind). A proof for Formula (1) in KeYmaera X uses the loop invariant accInv (see Appendix D). Automation of dL proofs as well as control envelope and invariant synthesis is discussed in the literature [57, 76, 78, 89].

**ModelPlex.** Many CPS safety properties can be formulated through a dL formula $\phi \rightarrow \big[(\alpha_{\text{ctrl}}; \alpha_{\text{plant}})^*\big]\psi$ where $\phi$ describes initial conditions, and $\psi$ describes the safety criterion to be guaranteed when following the control-plant loop. ModelPlex shielding [69] allows the synthesis of correct-by-construction *controller monitor formulas* $\zeta_c$ that ensure an implementation's runtime behavior matches the envelope $\alpha_{\text{ctrl}}$. Interpreting an implementation's action as a state transition and denoting the old state's variables as $x_i$ and the new state's variables as $x_i^+$, $\zeta_c$ tells us which combinations of $x_i$ and $x_i^+$ (i.e. which state transitions) are admissible w.r.t. $\alpha_{\text{ctrl}}$ (see Definition 4 in Appendix A).

**Running Example.** We can apply ModelPlex on the proven contract in Formula (1) to synthesize a monitor for $\alpha_{\text{ctrl}}$. For this simple scenario, the resulting controller monitor formula [1] tells us what new acceleration value $a_{\text{rel}}^+$ may be chosen given the current values of $p_{\text{rel}}, v_{\text{rel}}$:

$$\text{accCtrlFml} \equiv a_{\text{rel}}^+ = B \vee \big( a_{\text{rel}}^+ = 0 \wedge \text{accCtrl}_0^+ \big) \vee \big( -B \leq a_{\text{rel}}^+ \leq A \wedge \text{accCtrl}_1^+ \big). \quad (2)$$

Here, accCtrl$_i^+$ is the constraint accCtrl$_i$ with $a_{\text{rel}}$ replaced by $a_{\text{rel}}^+$. Given an action of a concrete controller implementation that changes $a_{\text{rel}}$ to $a_{\text{rel}}^+$, Formula (2) tells us if this action is in accordance with the strategy modeled by $\alpha_{\text{ctrl}}$, i.e. whether we have a proof of safety for the given state transition.

## 2.2 Neural Network Verification

This work focuses on feed-forward neural networks typically encountered in NNCSs. The behavior of an NN with input dimension $I \in \mathbb{N}$ and output dimension $O \in \mathbb{N}$ can be summarized as a function $g : \mathbb{R}^I \rightarrow \mathbb{R}^O$. The white-box behavior is described by a sequence of $L \in \mathbb{N}$ hidden layers with dimensions $n^{(k)}$ that iteratively transform an input vector $x^{(0)} \in \mathbb{R}^I$ into an output vector $x^{(L)} \in \mathbb{R}^O$. The computation of layer $k$ is given by $x^{(k+1)} = f^{(k)}\big(W^{(k)}x^{(k)} + b^{(k)}\big)$, i.e. an affine transformation (with FOL$_{\mathbb{N}\mathbb{R}}$ representable numbers) followed by a nonlinear activation function $f^{(k)}$. We distinguish different classes of NNs. To this end, we decompose the activation functions $f^{(k)}$ as $f^{(k)}(x) = \sum_{i=1}^s \mathbb{1}_{q_i}(x) f_i(x)$ where $f_i$ are functions, $q_i$ are formulas over $n^{(k)}$ variables and $\mathbb{1}_{q_i}(x)$ is 1 iff $q_i(x)$ is true and 0 otherwise. Table 2 summarizes which results are applicable to which NN class. Each class is a subset of the previous class, i.e. our theory (Section 3) is widely applicable while our implementation (Section 5) focuses on the most common NNs. Open-loop NNV tools analyze NNs in order to verify properties on input-output relations. Their common functionality is reflected in the

---

[1]The formula furthermore keeps the new $p_{\text{rel}}^+$ and $v_{\text{rel}}^+$ values unchanged ($p_{\text{rel}} = p_{\text{rel}}^+ \wedge v_{\text{rel}} = v_{\text{rel}}^+$), elided.

| NN class | All $f_i$ | All $q_i$ in | Applicable | Decidable | Example |
|---|---|---|---|---|---|
| piece-wise Noetherian | Noetherian | $FOL_{N\mathbb{R}}$ | Section 3 | | Sigmoid |
| piece-wise Polynomial | Polynomial | $FOL_{\mathbb{R}}$ | Section 3 | ✓ | $x^2$ |
| piece-wise Linear | Linear | $FOL_{L\mathbb{R}}$ | Sections 3 and 4 | ✓ | MaxPool |
| ReLU | $f^{(k)}(x) = \max(0, x)$ | | Sections 3 to 5 | ✓ | ReLU |

Table 2: Applicability of our results on NNCS safety and decidability of the safety verification problem: Each class is a subset of its predecessor in the table.

VNNLIB standard [15, 21]. Off-the-shelf tools are limited to linear, normalized queries (Definition 1). To address this challenge, we present a lifting procedure for the verification of generic (i.e. nonlinear and not normalized) open-loop NNV queries over polynomial real arithmetic (Section 4).

**Definition 1** (Open-Loop NNV Query). *An* open-loop NNV query *consists of a formula $p \in FOL_{\mathbb{R}}$ over free input variables $Z = \{z_1, \ldots, z_I\}$ and output variables $x_1^+, \ldots, x_O^+$. We call $p$* normalized *iff $p$ is a conjunction of some input constraints and a disjunctive normal form over mixed/output constraints, i.e. it has the structure $\bigwedge_j p_{1,j}(z_1, \ldots, z_I) \wedge \bigvee_{i \geq 2} \bigwedge_j p_{i,j}(z_1, \ldots, z_I, x_1^+, \ldots, x_O^+)$, where all $p_{i,j}$ are atomic real arithmetic formulas and all $p_{1,j}$ only contain the free variables from $Z$. We call a query* linear *iff $p \in FOL_{L\mathbb{R}}$ and call it* nonlinear *otherwise.*

## 3 VerSAILLE: Verifiably Safe AI via Logically Linked Envelopes

We introduce VerSAILLE, our approach for the verification of NNCSs via dL contracts. The key idea of VerSAILLE are *nondeterministic mirrors*, a mechanism that allows us to reflect a given NN $g$ and reason within and outside of dL simultaneously. This allows us to instrument open-loop NNV techniques to prove an NN specification outside of dL which *implies* the safety of a corresponding (mirrored) dL model describing the NNCS. Reconsider the ACC example (Section 1) for which we synthesized a controller monitor formula in Section 2. The remaining open question is the following:

> If we replace the control envelope $\alpha_{\text{ctrl}}$ by a given piece-wise Noetherian NN $g$, does the resulting system retain the same safety guarantees?

**Summary of VerSAILLE** The input for VerSAILLE is a proven dL contract. Additionally, one may provide an inductive invariant $\zeta_s$ to simplify the subsequent state space analysis. Using ModelPlex's synthesis of $\zeta_c$, VerSAILLE constructs a nonlinear open-loop NNV query. If we verify this query on an NN $g$, then the NNCS where we *substitute* the control envelope by $g$ retains the same safety guarantee.

**Running Example** Since one can only provide formal guarantees for something one can describe formally, we first need a semantics for what it means to substitute $\alpha_{\text{ctrl}}$ by $g$. To this end, we formalize a given piece-wise Noetherian NN $g$ as a hybrid program $\alpha_g$ which we call the *nondeterministic mirror* of $g$ (see Definition 16 and Lemma 17 in Appendix C). E.g. for ACC, this program must have two free (i.e. read) variables $p_{\text{rel}}, v_{\text{rel}}$ and one bound (i.e. written) variable $a_{\text{rel}}$ and must be designed in such a way that it exactly implements the NN $g$. Showing safety (see question above) is then equivalent to proving the following dL formula where $\alpha_{\text{plant}}$ describes ACC's physical dynamics:

$$\text{accInit} \rightarrow \left[\left(\alpha_g; \alpha_{\text{plant}}\right)^*\right] p_{\text{rel}} > 0. \tag{3}$$

Since NNs do not lend themselves well to interactive analysis, an *automatable* mechanism to prove formula (3) *outside* the dL calculus is desirable. As discussed in Section 2, we can prove the safety ($p_{\text{rel}} > 0$) of $\alpha_{\text{ctrl}}$ via dL. Thus, if we can show that all behavior of the nondeterministic mirror $\alpha_g$ is *already* modeled by $\alpha_{\text{ctrl}}$, the safety guarantee carries over from the envelope to $\alpha_g$. To show this refinement relation [65, 79], we instrument the controller monitor accCtrlFml in Formula (2). We verify that the NN $g$ satisfies the controller monitor formula accCtrlFml (i.e. we show that $g$'s input-output relation satisfies Formula (2)). If this is the case, $\alpha_g$'s behavior is modeled by our envelope $\alpha_{\text{ctrl}}$. In practice, it is unnecessary that the behavior of $\alpha_g$ is modeled by $\alpha_{\text{ctrl}}$ *everywhere* (e.g. we are not interested in states with $p_{\text{rel}} \leq 0$). It suffices to consider all states within the inductive invariant accInv of the envelope's system (Formula (1)) as those are precisely the states for which the guarantee on $\alpha_{\text{ctrl}}$ holds. Thus, we can prove Formula (3) by showing that $g$ satisfies the following specification for all inputs:

$$\text{accInv} \rightarrow \text{accCtrlFml} \tag{4}$$

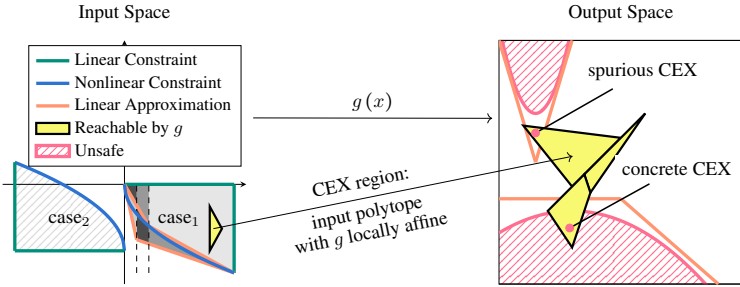

Figure 3: Visualization of the nonlinear verification algorithm Mosaic in Section 4

VerSAILLE also allows to soundly constrain the system to value ranges where the NN has been trained (Lemma 20). Verifying the queries generated by VerSAILLE with a (nonlinear) open-loop NNV tool (Definition 22) then implies safety of the NNCS (full formalism see Appendix C.1):

**Theorem 2** (Soundness). *Let $g$ be a piece-wise Noetherian NN. Further, let $C \equiv \left(\phi \to \left[(\alpha_{ctl}; \alpha_{plant})^*\right] \psi\right)$ be a valid contract with controller monitor $\zeta_c \in FOL_{\mathbb{R}}$ and inductive invariant $\zeta_s \in FOL_{\mathbb{R}}$. If a sound Nonlinear Neural Network Verifier returns* `unsat` *for the query $p \equiv (\zeta_s \land \neg\zeta_c)$ on $g$ then $\phi \to \left[(\alpha_g; \alpha_{plant})^*\right] \psi$ is valid for the nondet. mirror $\alpha_g$.*

If $g$ is piece-wise polynomial, Formula (4) is expressible in $FOL_{\mathbb{R}}$ and therefore its verification decidable (Lemma 24 which is a special case of [79]). In practice, we can be much more efficient than naively applying real arithmetic theory solvers by relying on open-loop NNV technologies to check the negated property accInv ∧ ¬accCtrlFml: If this property is unsatisfiable for a given NN $g$, then Formula (3) is valid. Off-the-shelf open-loop NNV tools are unable to reason about Formula (4) due to its nonlinearities and the non-normalized formula structure. Thus, our second contribution (Section 4) lifts open-loop NNV tools to the task of verifying nonlinear, non-normalized queries.

## 4 Mosaic: Nonlinear Open-Loop NN Verification

Since NNCSs usually exhibit nonlinear physical behavior, the verification property $\zeta_s \land \neg\zeta_c$ will be nonlinear as well. $\zeta_s \land \neg\zeta_c$ is also a formula of arbitrary structure and not a normalized open-loop NNV query (Definition 1). This is evident in the verification query for our running example (see Appendix D) or in Figure 3 where the green and blue constraints (left) describe two independent nonlinear input regions ($\text{case}_1$ and $\text{case}_2$) and the red regions (right) correspond to the unsafe regions for $\text{case}_1$. In contrast to this, off-the-shelf open-loop NNV tools (e.g. nnenum [12] or Marabou [59]) only support the verification of linear, normalized open-loop NNV queries on piece-wise linear NNs. Mosaic is a framework that allows us to *lift* off-the-shelf open-loop NNV tools for linear, normalized open-loop NNV queries to polynomial queries of arbitrary logical structure. This approach has the notable advantage that Mosaic's capabilities grow as open-loop NNV technology advances while retaining completeness w.r.t. polynomial constraints. An overview of the algorithm is given in Algorithm 1 and Figure 3 which is explained throughout this section with further details in Appendix B.

**MOSAIC.** To tackle formulas with arbitrary logical structure we could use DNNV [87] which implements a simple expansion algorithm or the standard formulation of DPLL(T). Consider the specification in Figure 4 where the x-axis represents the NN's inputs, the y-axis possible outputs, and the red regions are considered unsafe: DPLL(T), or an expansion algorithm, would enumerate all 7 red regions individually and then invoke an open-loop NNV tool for each. However, this becomes prohibitively inefficient when used for reachability-based open-loop NNV tools: Such tools will compute the reachable regions for $A_1$ three times, for $A_2$ three times, and for $A_3$ two times. Instead, we propose MOSAIC which first only enumerates *input* regions (e.g. the regions $A_1$, $A_2$, and $A_3$) and

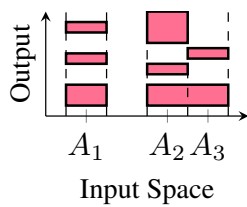

Figure 4: Enumeration of specification regions

then enumerates unsafe regions in the output space *per input region*. The unsafe regions are then aggregated into disjunctive normal form. Hence, for the example in Figure 4, MOSAIC would only yield *three* normalized queries which are still compatible with the standardized interface for NN

Algorithm 1: Verification of nonlinear queries on piece-wise linear NNs: ENUM internally uses off-the-shelf open-loop NNV tools for the verification of linear normalized queries on NNs.

---

**Input:** Formula $p \in \text{FOL}_{\mathbb{R}}$, ranges $R$, piece-wise linear NN $g$

   $p_o \leftarrow \text{LINEARIZE}(p, R)$                             $\triangleright$ Generate linearized $p$

   **for all** $(q_l, q_n) \in \text{MOSAIC}(p_o)$ **do**               $\triangleright$ Iterate over normalized queries

       **for all** $(\iota, \omega) \in \text{ENUM}(g, q_l)$ **do**        $\triangleright$ uses GENERALIZE & open-loop NNV

           **if** $\text{FILTER}(q_l \wedge q_n, \iota, \omega)$=concrete **then return** unsafe

   **return** safe                                $\triangleright$ No concrete counterexamples found

---

verification tools [15]. We call this partitioning of the input space a *mosaic* and, in reminiscence of this analogy, we call the individual queries *azulejos* ([ɑ.θu'le.xo], see e.g. regions in shades of gray in Figure 3). MOSAIC guarantees that reachability-based open-loop NNV tools do not explore the same input region multiple times (see *flatness* result in Proposition 10; Appendix B). For our ACAS case study, naive rewriting (as done by e.g. DNNV [87]) may produce up to 39 trillion propositionally feasible queries. In contrast, MOSAIC only produces 19k queries (see Table 7 in Appendix E.2). MOSAIC also separates nonlinear constraints $q_n$ (must be checked outside open-loop NNV) from linear constraints $q_l$ (can be passed to open-loop NNV).

On the technical side, Mosaic proceeds by executing a SAT solving based DPLL(T) loop until a satisfiable conjunction of *input* constraints is found. At this stage, we fix the conjunction's linear input constraints (i.e. the azulejo) and an inner loop enumerates conjunctions over mixed/output constraints that are satisfiable in combination with the fixed azulejo. For each such conjunction, we save the conjunction of linear mixed/output constraints. This results in a linear, normalized Open-Loop NNV query (conjunction over input, disjunctive normal from over output). We employ a similar inner loop to enumerate satisfiable conjunctions of nonlinear constraints to later check counterexamples via SMT solving (see Retaining completeness). At each step, we interleave propositional and theory solving to discard conjunctions unsatisfiable in real arithmetic as early as possible.

**LINEARIZE.** To verify polynomial specifications via off-the-shelf (linear) open-loop NNV, we need to soundly approximate the query's nonlinear constraints. In principle, we could perform this approximation for each azulejo (see MOSAIC) separately. To this end, consider an atomic polynomial constraint $p(x) \leq 0$ which is part of a query. If there are two azulejos with the constraints $p(x) \leq 0$ and $p(x) > 0$, separate approximation would lead to the constraint $\overline{p}(x) \leq 0$ and $\underline{p}(x) > 0$ (with $\overline{p}/\underline{p}$ resp. over/under-approximations of $p$). This would *duplicate* the exploration of the area in between $\overline{p}$ and $\underline{p}$ and can lead to an exponential blowup for many approximations. Instead, we use a *global* piece-wise approximation via OVERT [88] (orange lines around the blue nonlinear constraint on Figure 3) and integrate the approximate constraints into the original query via implications (e.g. we add $\overline{p}(x) \leq 0 \rightarrow p(x) \leq 0$). These additional linear constraints are then automatically enumerated via MOSAIC (see azulejos for $\text{case}_1$ in Figure 3). The linearization happens for input as well as output constraints. When passing the open-loop NNV query to the off-the-shelf tool, we soundly omit the nonlinear constraints, thus only leaving behind linear constraints $q_l$ (green and orange in Figure 3).

**Retaining completeness.** Without further efforts, MOSAIC and LINEARIZE together yield a sound algorithm to check nonlinear, not normalized open-loop NNV queries, but not a complete one: A reachability analysis via open-loop NNV for a given azulejo may produce spurious counterexamples that are an artifact of the linearization (see Figure 3) The key insight to achieve completeness is the observation that for piece-wise *linear* NNs any concrete counterexample generated via open-loop NNV corresponds to a *counterexample region* $\iota$ (yellow polytope on the left of Figure 3) on which the NN reduces to an affine transformation $\omega$ (obtained by fixing piece-wise functions to the linear segment of the concrete counterexample, i.e. we fix the value of the $\mathbb{1}$ functions; see also Section 2.2). This insight can be used in two ways: First, we can use it to enumerate all counterexample regions $\iota$ (by adjusting the tool's internal enumeration and/or via the algorithm ENUM; see Appendix B.3). Secondly, for a given counterexample region, we can check whether there exists a concrete counterexample to the nonlinear specification via SMT solving (in Algorithm 1 this is performed by FILTER; see Appendix B.3). By exploiting the affine transformation $\omega$, our SMT encoding for nonlinear constraints has at most $I$ free variables (for $I$ input dimensions of the NN) and is therefore *significantly more tractable* than an encoding of the entire NN in SMT.

Using the components outlined above (for details see Appendix B), we prove the soundness and completeness of Mosaic. Completeness turns out to also be of *practical relevance* as approximation alone would have failed to verify the DNC NN of the ACAS benchmark discussed in Section 5.

**Theorem 3** (Soundness and Completeness). *Let $g$ be a piece-wise linear NN, $p$ be a real arithmetic formula and $R$ variable ranges for all in- and output variables of $g$. Algorithm 1 returns* `unsafe` *iff there exists an input $z \in \mathbb{R}^I$ such that $(z, g(z))$ is in the range $R$ and $p(z, g(z))$ is satisfied.*

In Section 5 we build upon nnenum [10,12] which can enumerate all counterexample regions of an NN with $N$ ReLU activations in time $\mathcal{O}(2^N)$, and upon cylindrical algebraic decomposition (CAD [28]) with complexity $\mathcal{O}(2^{2^V})$ for $V$ variables[2]. Assuming $M$ atomic formulas in the open-loop NNV query and $I$ input dimensions this yields a worst-case runtime of $\mathcal{O}(2^{M+\mathbf{N}+2^I})$. This is an exponential improvement over naive $\mathcal{O}(2^{M+2^{\mathbf{N}+I}})$ CAD encodings as $N >> I$. In practice the performance is even better; usually, MOSAIC explores fewer queries, nnenum returns less counterexample regions and SMT solving tends to perform very well for the small input dimensions $I$ of NNCS control.

## 5 Evaluation

We implemented Mosaic for ReLU NNs in a new Julia [16] tool called N[3]V based on the software packages nnenum [10, 12], PicoSAT [17, 19] and Z3 [32, 52]. We provide wall-clock times on an AMD Ryzen 7 PRO 5850U CPU (N[3]V is sequential; nnenum uses multithreading). The evaluation presented in this section focuses on vertical airborne collision avoidance (VCAS), with other experiments in Appendix E. Airborne Collision Avoidance Systems try to recognize plane trajectories that might lead to a Near Mid-Air Collision (NMAC) with other aircrafts and advise the pilot to avoid such collisions. NMACs are defined as two planes (ownship and intruder) flying closer than 500 ft horizontally or 100 ft vertically. Currently, the Federal Aviation Administration (FAA) develops a new airborne collision avoidance system called Airborne Collision Avoidance System X (ACAS X) [71]. Prior work by Jeannin et al. [50] showed a nondeterministic, provably safe dL envelope for airborne collision avoidance. While the original proposal for ACAS X [45, 60] was shown to be unsafe [50], the correctness of a VCAS NN implementation [53, 55] was proven [54] and disproven [56] in special cases. The proposed NNs contain 6 hidden layers with 45 neurons each and produce one of 9 collision avoidance advisories (*Strengthen Climb to at least 2500 ft/min* (SCL2500) to *Strengthen Descent to at least 2500ft/min* (SDES2500); see Table 9 in Appendix F for a list of allowed advisories). The objective of the NNs is to ensure safety while minimizing the number of alerts sent to the pilot. We present an *exhaustive* analysis of the VCAS NNCS for level flight intruders.

We provide additional experimental results in Appendix E. First, we demonstrate the feasibility of our approach for the running example of ACC. Depending on NN size and chosen linearization, our approach can verify or exhaustively enumerate counterexamples for the NNCS in 47 to 300 seconds. For a case study on Zeppelin steering under (uniformly sampled) wind perturbations, we adapt a differential hybrid games formalization [77] to analyze an NN controller trained by us. Here, we encountered a controller that showed

| Prev. Adv. | Status | Time | CE regions | First CE |
|---|---|---|---|---|
| DNC | **safe** | 0.35 h | — | — |
| DND | **safe** | 0.28 h | — | — |
| DES1500 | **unsafe** | 5.45 h | 49,428 | 0.04 h |
| CL1500 | **unsafe** | 5.18 h | 34,658 | 0.08 h |
| SDES1500 | **unsafe** | 4.05 h | 5,360 | 0.97 h |
| SCL1500 | **unsafe** | 4.89 h | 11,323 | 0.36 h |
| SDES2500 | **unsafe** | 3.66 h | 5,259 | 1.39 h |
| SCL2500 | **unsafe** | 4.45 h | 7,846 | 0.53 h |

Table 3: Verification of ACAS NNs for level flight: Previous advisory (=Prev. Adv.), runtime; number of counterexample (=CE) regions and time to the discovery of the first CE.

very positive empirical performance while being provably unsafe in large parts of the input space: While performing very well on average, the control policy was vulnerable to unlikely wind perturbations – an issue we only found through our verification. For ACC, we also perform a comparison to other techniques: While Closed-Loop techniques are useful for the analysis of bounded-time safety, their efficiency greatly depends on the system's dynamics and the considered input space. Our

---

[2]For simplicity of analysis we fix the complexity's other variables (e.g. maximal degree).

[3]See `https://github.com/samysweb/NCubeV` or [91]

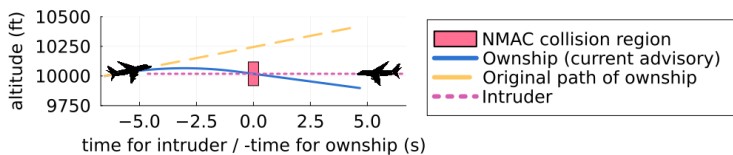

Figure 5: An unsafe advisory by the Airborne Collision Avoidance NN: After a previous advisory to climb at least 1500ft/min, the NN advises to reverse vertical direction ("Strengthen Descent to at least 1,500 ft/min"). This leads to an NMAC 6 seconds later. More examples are in Appendix G.

infinite-time horizon approach can be more efficient than Closed-Loop techniques as it evades the necessity to analyze the system's dynamics along with the NN (see Table 6). Usually, it is desirable to show infinite-time safety on the entire (controllable) state space. However, the approximation errors incurred via prior closed-loop NNV techniques prohibit this as they will either ignore states inside the controllable region or allow unsafe actions pushing the system outside its controllable region. Conversely, SMT-based techniques do not have these approximation issues, but cannot scale to NNs of the size analyzed in this work. We also provide a conceptual comparison demonstrating the efficiency of the Mosaic procedure for normalized query generation over DNNV's expansion-based algorithm (see Table 7), naive SMT solving (Table 8) and Genin et al.'s tailored ACAS approach [42].

**Verification Results for VCAS.** We use the nondeterministic control envelope and loop invariant by Jeannin et al. [50, Thm. 1] to analyze safety for intruders in level flight (i.e. intruder vertical velocity is 0). For the same reasons as prior work [50], we ignore Clear-of-Conflict advisories. The open-loop NNV queries obtained via VerSAILLE had up to 112 distinct atoms and trees up to depth 9. To determine the maximal output neuron, open-loop NNV queries initially contain atoms sorting the NN's outputs $x_1, \ldots, x_n$ (e.g., $x_1 \leq x_2$). To avoid enumerating all permutations, we perform symmetry elimination via an atomic predicate encoding that some output $i$ is maximal. We analyzed the *full range* of possible NN inputs for intruders in level flight (relative height $|h| \leq 8000$ft, ownship velocity $|v| \leq 100$ft/s and time to NMAC $6s \leq \tau \leq 40s$). Our results are in Table 3: **safe** implies that the NN's advisories (other than Clear-of-Conflict) in this scenario *never* lead to a collision when starting within the invariant. The safety for DNC was only verifiable through SMT filtering (approximation yielded spurious counterexamples). This *underscores the importance of Mosaic's completeness* and implies that [56] is insufficient to prove safety. A non-exhaustive analysis for non-level flight yielded counterexamples even for DNC/DND (see Appendix G). For unsafe level flight scenarios, we exhaustively characterize unsafe regions. This characterization goes far beyond characterizations in prior work [56] that were generated using manual approximation and resulted in (non-exhaustive) point-wise characterizations. Figure 5 shows a concrete avoidable NMAC (more examples are in Appendix G).

**Scalability.** $N^3V$ provides guarantees for an NN's full input space. Hence, $N^3V$ is not directly comparable to, e.g., the scalability of local robustness verifiers, that, while sometimes scaling to hundred thousands of ReLUs [20], only analyze tiny fractions of the input space [92, Sec. 6]. Contemporary work on global properties outside NNCSs has been scaled to 100 ReLU nodes [8]. Related work on infinite-time NNCSs (see also Section 6) has not been scaled beyond 30 nodes in similar settings [25, 64]. The largest NNs verified by us so far had 270 ReLUs, indicating $N^3V$ is at the frontier of State-of-the-Art scalability for *global* properties. Some NNCS applications (e.g. ACAS [55]) turn to NNs to efficiently encode complex strategies in mid-scale NNs. $N^3V$ scales to NNs of this kind.

## 6 Related Work

**Shielding.** Justified Speculative Control [39] is closely related in its use of dL. However, we verify the NNCS *a priori* instead of treating ML models as a black box and *a posteriori* using runtime enforcement techniques [61, 69]. Shielding can also be used to train models such that they (probabilistically) conform to a shield/monitor [6]. Training methodologies are beyond the scope of this paper.

**Barrier Certificates.** An orthogonal direction of research explores learning Neural Barrier/Lyapunov Functions to prove safety properties [30]. Although initially used for "pure" dynamical and hybrid systems (without NNs) [1, 2, 4, 70, 73, 83, 101, 104], the methods have since been extended to NNCS with discrete [9, 26, 64] and continuous [25, 31, 81] time behavior. While the former works

can only approximate continuous time behavior, the latter techniques use off-the-shelf SMT solvers (see also Appendix E.2) for certificate verification which severely limits scalability. While some works ignore verification entirely [31, 81], the remaining work only considered *linear single-layer* NNs [25, Appendix] verified with dReal (see SMT comparison in Appendix E.2). Dawson et al. [31] note "scalable verification for learned certificate functions remains an open problem". Using $N^3V$ as an alternative to SMT for certificate verification is future work. VerSAILLE evades the necessity to *learn* barrier certificates by reusing established control-theory literature.

**Open-Loop NNV.**   VEHICLE [29] integrates open-loop NNV with Agda. However, VEHICLE only allows importing normalized, linear properties which limits applicability to CPS verification in realistic settings. Open-loop NNV tools [12, 23, 35, 43, 44, 58, 59, 95, 98–100] do not consider the physical environment and thus *cannot* guarantee the safety of an NNCS. Even in cases where such methods allow nonlinear behavior in activation functions, they do not admit the verification of arbitrary polynomial constraints over the input and output space. Most methodologies could be integrated into Mosaic's framework, i.e. we can lift complete off-the-shelf open-loop NNV tools to verify polynomial constraints with arbitrary structure. DNNV [87] proposed an approach for open-loop NNV query normalization using a simple expansion algorithm. DNNV has the same limitations as all open-loop NNV tools (no NNCS analysis; no nonlinear constraints) and is less efficient than Mosaic w.r.t. NN reachability analysis (see Section 4). Pre-image computation [62, 68, 103] computes input regions producing a fixed NN prediction. For efficiency, our work constrains the input space with invariants and value ranges—this efficiency would be lost by a backward computation alone.

**Related Techniques.**   Unlike [13, 80], we support *arbitrary* polynomial constraints and retain completeness. Moreover, these works do not support arbitrary logical structure and represent open-loop NNV techniques unable to analyze NNCSs. Some prior work used techniques for constructing counterexample regions [33, 102] but only for *individual* datapoints—neither to compute exhaustive characterizations nor to regain completeness for incomplete verifiers. Unlike classical DPLL(T) [40], MOSAIC is tailored to theory-solving w.r.t. reachability analyzers. To this end, MOSAIC groups output regions with the same input constraints which deduplicates work (see Section 4).

**Closed-Loop NNV.**   Closed-loop NNV tools [5, 18, 34, 37, 47–49, 85, 88, 93, 94, 97] only consider a fixed time horizon and thus cannot guarantee infinite-time horizon safety (see Appendix E.2). Unlike [11, 42, 56], our approach is automated and applicable to *any* CPS expressible in dL (not just ACAS X; see case studies in Appendix E). Other approaches verify simplified control outputs [42]; rely on hand-crafted approximations while lacking exhaustive counterexamples characterizations [56]; or require quantization effectively analyzing only a surrogate system instead [11].

# 7   Conclusion and Future Work

This work presents VerSAILLE, the first technique exploiting dL contracts to prove safety of NNCSs with piece-wise Noetherian NNs. VerSAILLE requires open-loop NNV tools capable of verifying non-normalized polynomial properties that did not exist. Thus, with Mosaic we present an efficient, *sound and complete* approach for the verification of such properties on piece-wise linear NNs. We implemented Mosaic for ReLU NNs in the tool $N^3V$ and demonstrate the applicability and scalability of our approach on multiple case studies (Section 5 and Appendix E). The application to NNCSs by Julian et al. [53, 55] shows that our approach scales even to intricate, high-stakes applications such as airborne collision avoidance. Our results underscore the categorical difference of our approach to closed-loop NNV techniques. Overall, we demonstrate an efficient and generally applicable approach that opens the door for developing of goal-oriented *and* infinite-time horizon safe NNCSs in the real world.

**Future Work.**   We believe there is potential for engineering and algorithmic improvements that could further improve the performance of $N^3V$. Our implementation could also be extended to other piece-wise linear activation functions. We would also like to explore the efficiency of $N^3V$ for non-polynomial specifications w.r.t. suitable SMT solvers. Mosaic may be of interest even beyond NNCS verification in VerSAILLE. Since, e.g., neural barrier certificate verification also requires nonlinear open-loop NNV, Mosaic could be equally applicable in this context. Finally, it would be interesting to apply our approach to further case studies. To this end, the bottle-neck is currently the limited availability of NNCS which are safe w.r.t. an infinite-time horizon.

## Acknowledgements

This work was supported by funding from the pilot program Core-Informatics of the Helmholtz Association (HGF) and by an Alexander von Humboldt Professorship. Part of this research was carried out while Samuel Teuber was funded by a scholarship from the International Center for Advanced Communication Technologies (interACT) in cooperation with the Baden Württemberg Foundation.

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

# A    Additional Background

This section provides additional background on Differential Dynamic Logic and the status of various activation functions w.r.t. our classification in Table 2.

## A.1    Differential Dynamic Logic

Differential dynamic logic (dL) [74–76, 78] can analyze models of hybrid systems that are described through *hybrid programs*. The syntax of hybrid programs with Noetherian functions is defined by the following grammar, where the term $e$ and formula $Q$ are over real arithmetic with Noetherian functions:

$$\alpha, \beta \ ::= \ x := e \mid x := * \mid ?Q \mid x' = f(x)\&Q \mid \alpha \cup \beta \mid \alpha; \beta \mid \alpha^* \tag{5}$$

The semantics of hybrid programs are defined by a transition relation over states in the set $\mathcal{S}$, each assigning real values to all variables. For example, the assignment state transition relation is defined as $[\![x := e]\!] = \{(\nu, \omega) \in \mathcal{S}^2 \mid \omega = \nu_x^{\nu(e)}\}$ where $\nu_x^{\nu(e)}$ denotes the state that is equal to $\nu$ everywhere except for the value of $x$, which is modified to $\nu(e)$. The other programs in the same order as in (5) describe nondeterministic assignment of $x$, test of a predicate $Q$, continuous evolution along the differential equation within domain $Q$, nondeterministic choice, sequential composition, and nondeterministic repetition. For a given program $\alpha$, we distinguish between bound variables $BV(\alpha)$ and free variables $FV(\alpha)$ where bound variables are (potentially) written to, and free variables are read. The formula $[\alpha]\psi$ expresses that $\psi$ is always satisfied after the execution of $\alpha$ and $\langle\alpha\rangle\psi$ that there exists a state satisfying $\psi$ after the execution of $\alpha$. If a state $\nu$ satisfies $\psi$ we denote this as $\nu \vDash \psi$. dL comes with a sound and relatively complete proof calculus [74, 76, 78] as well as the interactive theorem prover KeYmaera X [38].

**ModelPlex.**    As demonstrated in the example, many safety properties for CPSs can be formulated through a dL formula with a loop in which $\alpha_{\text{ctrl}}$ describes the (discrete) software, and $\alpha_{\text{plant}}$ describes the (continuous) physical environment:

$$\phi \rightarrow \left[\left(\alpha_{\text{ctrl}}; \alpha_{\text{plant}}\right)^*\right]\psi. \tag{6}$$

$\phi$ describes initial conditions, and $\psi$ describes the safety criterion to be guaranteed when following the control-plant loop. To ensure that the behavior of controllers and plants in practice match all assumptions represented in the contract, ModelPlex shielding [69] synthesizes correct-by-construction monitors for CPSs. ModelPlex can also synthesize a correct controller monitor formula $\zeta_c$ (Definition 4). The formula $\zeta_c$ encodes a relation between two states. If $\zeta_c$ is satisfied, then the variable change from $x_i$ to $x_i^+$ corresponds to behavior modeled by $\alpha_{\text{ctrl}}$, i.e. the change upholds the guarantee from Formula (6). To reason about this state relation, we say that a state tuple $(\nu, \omega)$ satisfies a formula $\zeta$ (denoted as $(\nu, \omega) \vDash \zeta$) iff $\nu^{\omega(x_1)...\omega(x_n)}_{x_1^+...x_n^+} \vDash \zeta$ (i.e. $\nu$ with the new state's value $\omega(x_i)$ as the value of $x_i^+$ for all $i$, satisfies $\zeta$).

**Definition 4** (Correct Controller Monitor [69]). *A controller monitor formula $\zeta_c$ with free variables $x_1, x_1^+, \ldots, x_n, x_n^+$ is called* correct *for the hybrid program controller $\alpha_{ctrl}$ with bound variables $x_1, \ldots, x_n$ iff the following dL formula is valid: $\zeta_c \rightarrow \langle\alpha_{ctrl}\rangle \bigwedge_{i=1}^n x_i = x_i^+$.*

## A.2    Classification of Activation Functions

Table 4 is meant to provide a brief overview on the status of various common activation functions w.r.t. the classes from Table 2. The chosen activation functions are representative examples from the comprehensive survey by Kunc et al. [63]. For piece-wise polynomial and Noetherian functions, any activation function represented by a polynomial over the activation functions presented in Table 4 is resp. piece-wise polynomial or Noetherian. For piece-wise linear functions, any linear combination is resp. piece-wise linear. This table does not claim to be comprehensive—many more activation functions can be classified into one of the categories.

# B    Mosaic

An overview of the algorithm is given in Figure 3: A piece-wise linear NN $g$ is a function which maps from an input space (left) to an output space (right). We consider a part of the input space that

| Activation Function | Justification |
|---|---|
| **ReLU NNs** | |
| ReLU | ReLU |
| **piece-wise linear NNs** | |
| LeakyReLU [63, 3.6.2] | Split into $< 0$ and $\geq 0$. |
| HardTanh [63, 3.6.18] | Split into 3 regions. |
| **piece-wise polynomial NNs** | |
| Square-based activation functions [63, 3.8] | by definition. |
| Polynomial universal activation function [63, 3.16] | by definition for integer exponents. |
| **piece-wise Noetherian NNs** | |
| Sigmoid [63, 3.2] | $\sigma'(x) = \sigma(x)(1 - \sigma(x))$ |
| Tanh [63, 3.2] | $\tanh(x) = 2\sigma(2x) - 1$ |
| Arctan [63, 3.2.4] | $\arctan'(x) = 1/(1 + x^2)$ |
| SiLU [63, 3.3] | Polynomial over $\sigma$ |
| $\exp(x)$ | $\exp'(x) = \exp(x)$ |
| GELU [63, 3.3.1] | In a Noetherian Chain with $\exp$ and the gaussian error funciton |
| Approximate GELU [63, 3.3.1] | Polynomial over $\sigma$ or $\tanh$ |
| Softmax [63, 3.5][a] | Polynomial over $\exp$. |

Table 4: Overview on classification of activation functions.

[a]The analysis of softmax can often be avoided by using the observation that its application does not change the order of outcomes, i.e. for classification-like tasks we can use the maximum before Softmax application instead.

is constrained by linear (orange) and nonlinear constraints (blue). As our query is not normalized, it may talk about multiple parts of the input space, e.g. in our case the two sets labeled with $case_1$ and $case_2$. For any such part of the input space, say $case_1$, we have a specification about *unsafe* parts of the output space which must not be entered (red dashed areas on the right). For classical open-loop NNV the task is then, given a single input polytope, to compute the set of reachable outputs for $g$ and to check whether there exists an output reaching an unsafe output polytope. In our case the task is more complicated, because the input is not a polytope, but an arbitrary polynomial constraint. Moreover, for each polynomial input constraint ($case_1$ and $case_2$ in Figure 3) we may have different nonlinear unsafe output sets. In order to retain soundness, we over- *and* underapproximate nonlinear constraints (see turquoise linear approximations around the blue and red nonlinear constraints in Figure 3). Once all nonlinear queries have approximations, we generate a *mosaic* of the input space where each *azulejo* (i.e. each input region) has its own normalized open-loop NNV-query (polytope over the input; disjunction of polytopes over the output). We must not only split between the two original cases ($case_1$ and $case_2$), but also between different segments of the approximating constraints (see the polytopes on the left in four shades of gray). Each normalized query has associated nonlinear constraints that must be satisfied, but cannot be checked via off-the-shelf open-loop NNV tools. Using the normalized, linear open-loop NNV queries we then instrument off-the-shelf tools to check whether any overapproximated unsafe region (turquoise on the right) is reachable. The amber colored regions represent parts of the outputs reachable by $g$: As can be seen by the two red dots, a reachable point within the overapproximated unsafe region may be a concrete unsafe output, or it may be spuriously unsafe due to the overapproximation. To retain completeness, we need to exhaustively filter spurious counterexamples. This is achieved by *generalizing* the counterexample to a region around the point in which the behavior of $g$ is equivalent to a single affine transformation. Such counterexample regions always exist due to $g$'s piece-wise linearity. For example, in Figure 3 $g$ affinely maps the input space triangle (left) to the upper output space triangle (in amber on the right). We can then check for concrete counterexamples in this region w.r.t. the affine transformation using an SMT solver. This avoids the need to encode the entire NN in an SMT formula. By excluding explored regions, we can then enumerate all counterexample regions and thus characterize the unsafe input set.

The approach is outlined in Algorithm 1 and proceeds in four steps, all of which will be presented in detail throughout the sections below: First, LINEARIZE generates approximate linearized versions for all nonlinear atoms of $p$ on a bounded domain and enriches the formula with these constraints ($p_o$). Next, MOSAIC generates a mosaic of $p_o$'s input space where each azulejo (i.e. each input space region)

has an associated linear normalized query $q_l$. Each $q_l$ is paired with an associated disjunctive normal form of nonlinear constraints $q_n$. The disjunction over all $q_l \wedge q_n$ is equivalent to the input query $p_o$ and the disjunction over all $q_l$ overapproximates LINEARIZE's input $p$. Each of the linear queries $q_l$ is processed by ENUM which internally uses an off-the-shelf open-loop NNV tool to enumerate all counterexample regions for a given query. Each counterexample region is defined through a polytope in the input space $\iota \subset \mathbb{R}^I$ and an affine mapping to the output space $\omega : \mathbb{R}^I \to \mathbb{R}^O$ that summarizes the NN's local behavior in $\iota$. The procedure FILTER then checks whether a counterexample region is spurious using an SMT solver. This task is easier than searching nonlinear counterexamples directly since the NN's behavior is summarized by the affine mapping $\omega$. Using the definitions from the following subsections, this procedure is sound and complete (see proof on page 28):

## B.1 Linearization

The procedure LINEARIZE enriches each nonlinear atom $a_i$ of an open-loop NNV query with linear approximations. The approximations are always with respect to a value range $R$ and we use overapproximations $\overline{a_i}$ (for any state $\nu$ with $\nu \vDash a_i \wedge R$ it holds that $\nu \vDash \overline{a_i}$) as well as underapproximations $\underline{a_i}$ (for any state $\nu$ with $\nu \vDash \underline{a_i} \wedge R$ it holds that $\nu \vDash a_i$). Essential to this component is the idea that LINEARIZE produces an equivalent formula: Approximate atoms do not replace, but complement the nonlinear atoms and the generation of concrete linear regions is left to the mosaic step (see Appendix B.2). LINEARIZE is defined as follows:

**Definition 5** (Linearization). LINEARIZE *receives an open-loop NNV query $p$ with nonlinear atoms $a_1, \ldots, a_k$ and value range $R$ s.t. $p \to R$ is valid. It returns a query $p \wedge \bigwedge_{i=1}^{k} \left( (a_i \to \overline{a_i}) \wedge (\underline{a_i} \to a_i) \right)$ where $\overline{a_i} \in FOL_{LR}$ (resp. $\underline{a_i} \in FOL_{LR}$) are overapproximations (resp. underapproximations) of $a_i$ w.r.t. $R$.*

We use an approximation procedure based on OVERT [88] while further approximating $\max / \min$ terms within OVERT. This results in a disjunction of linear constraints (see below for details). As highlighted above, LINEARIZE produces equivalent formulas and therefore retains the relations between linear and nonlinear atoms (see proof on page 28):

**Lemma 6** (Equivalence of Linearization). *Let $p \in FOL_{\mathbb{R}}$ be some open-loop NNV query and $p_o$ be the result of LINEARIZE$(p)$. Then $p$ is equivalent to $p_o$.*

**Approximation.** For concicenes we present our approximation approach for over-approximations. Our under-approximations are computed in the same manner, however lower and upper bound computation of terms is flipped in this case. We can approach the question of overapproximation construction from a perspective of models: For a given formula $\zeta$, let $[\![\zeta]\!] = \{\nu \in \mathcal{S} | \nu \vDash \zeta\}$ be the set of models (i.e. states satisfying $\zeta$). We then obtain the following Lemma for the relation between overapproximations and model sets:

**Lemma 7** (Supersets are Overapproximations). *Assume bounds $B$ on all variables and a formula $\zeta$. Another formula $\zeta_o \in FOL_{LR}$ is a linear overapproximation of $\zeta$ iff $[\![B \wedge \zeta]\!] \subseteq [\![B \wedge \zeta_o]\!]$.*

This presentation only considers the case of a polynomial constraint $\theta > 0$. Our approximation procedure begins by computing the relational approximation of $\theta$ using the OVERT algorithm [88]. By resolving intermediate variables introduced through OVERT, we obtain an approximation of the form $\underline{\theta_{pwl}} \leq \theta \leq \overline{\theta_{pwl}}$ where both bounds are piece-wise linear functions (i.e. linear real arithmetic with the addition of $\max$ and $\min$ operators). It then holds that $[\![B \wedge \theta > 0]\!] \subseteq [\![B \wedge \overline{\theta_{pwl}} > 0]\!]$. We now distinguish between univariate and multivariate piece-wise linear behavior: For univariate piece-wise linear behavior there is some variable $v \in \mathrm{V}(\overline{\theta_{pwl}})$ and some coefficient $c \in \mathbb{R}$ with terms $\theta_1, \theta_2$ such that $[\![B \wedge \overline{\theta_{pwl}} > 0]\!] = [\![B \wedge \overline{\theta_1} > 0 \wedge v > c]\!] \cup [\![B \wedge \overline{\theta_2} > 0 \wedge v \leq c]\!]$. In order to subsume piece-wise linear splits along variable $v$ which are close to $c$, we construct the following overapproximation for a small $\varepsilon > 0$ resulting in two normalized queries:

$$[\![B \wedge \overline{\theta_{pwl}} > 0]\!] \subseteq [\![B \wedge \overline{\theta_1} > 0 \wedge v > (c - \varepsilon)]\!] \cup [\![B \wedge \overline{\theta_2} > 0 \wedge v \leq (c + \varepsilon)]\!] .$$

For the multivariate cases we approximate the piece-wise linear behavior. In particular, we introduce a (to the best of our knowledge) novel, closed-form upper bound for the linear approximation of $\max$ terms:

**Lemma 8** (Upper Bound for multivariate max). *Let $f, g : \mathbb{R}^{I+O} \to \mathbb{R}$ be two linear functions, and let $B \subset \mathbb{R}^{I+O}$ be a closed interval box, then: Assume $x_g \coloneqq \arg\max_{x \in B} g(x) - f(x)$ and $x_f \coloneqq \arg\max_{x \in B} f(x) - g(x)$ where $f(x_f) - g(x_f)$ and $g(x_g) - f(x_f)$ are both positive. Further, assume the following assignments with $\gamma \coloneqq f(x_f) - f(x_g) - g(x_f) + g(x_g)$:*

$$\mu \coloneqq -\frac{g(x_f) - f(x_f)}{\gamma} \qquad\qquad c \coloneqq -\frac{(f(x_f) - g(x_g))(f(x_g) - g(x_g))}{\gamma}.$$

*In this case, it holds for all $x \in B$ that: $\mu f(x) + (1-\mu) g(x) + c \geq \max(f(x), g(x))$. In particular, it holds that*

$$\llbracket B \wedge (\max(f(x), g(x)) > 0) \rrbracket \subseteq \llbracket B \wedge ((\mu f(x) + (1 - \mu) g(x) + c) > 0) \rrbracket$$

*Proof.* At first, the choices for $\mu$ and $c$ may seem arbitrary, however they are actually the solution of the following set of equations:

$$\mu f(x_f) + (1 - \mu) g(x_f) + c = f(x_f)$$
$$\mu f(x_g) + (1 - \mu) g(x_g) + c = g(x_g)$$

The choice of $\mu$ and $c$ ensures that we obtain a shifted convex mixture of the two linear functions that matches $f$ and $g$ at their points of maximal deviation. We can now prove that this shifted mixture is indeed larger than $f$ or $g$ at any point within $B$. Let us begin by proving that our bound is larger than $g$ for $x \in B$. In the following each formula implies the validity of the formula above:

$$g(x) \leq \mu f(x) + (1 - \mu) g(x) + c$$
$$0 \leq \mu(f(x) - g(x)) + c$$
$$0 \leq -\frac{1}{\gamma}\left((g(x_f) - f(x_f))\underbrace{(f(x) - g(x))}_{\leq f(x_f) - g(x_f) \text{ for } x \in B} + (f(x_f) - g(x_g))(f(x_g) - g(x_g))\right)$$
$$0 \leq -\frac{1}{\gamma}\left((g(x_f) - f(x_f))(f(x_f) - g(x_f)) + (f(x_f) - g(x_g))(f(x_g) - g(x_g))\right)$$
$$0 \leq \frac{1}{\gamma}(f(x_f) - g(x_f))\gamma \Leftrightarrow g(x_f) \leq f(x_f)$$

$g(x_f) \leq f(x_f)$ is trivially true since $x_f$ was specifically chosen this way. We also have to prove that our bound is bigger than $f$ for $x \in B$:

$$f(x) \leq \mu f(x) + (1 - \mu) g(x) + c$$
$$0 \leq \underbrace{(g(x) - f(x))}_{\geq g(x_f) - f(x_f) \text{ for } x \in B} + \underbrace{\mu(f(x) - g(x)) + c}_{\geq f(x_f) - g(x_f) \text{ (see previous proof)}}$$
$$0 \leq g(x_f) - f(x_f) + f(x_f) - g(x_f) = 0$$

Thus we obtain an upper bound for the function. $\qquad\square$

By applying OVERT followed by the univariate resolution and multivariate overapproximation up to saturation, we compute an overapproximation and underapproximation for each nonlinear atom. Subsequently, we append these new formulas to the original formula as defined in Definition 5.

**Running Example.** The turquoise constraints in Figure 3 visualize exemplary linearized constraints. For ACC one nonlinear atom is $p_{\text{rel}} - \frac{v_{\text{rel}}^2}{2B} \geq 0$. The formula accApprox $\equiv p_{\text{rel}} - \frac{100^2}{2B} \geq 0 \wedge v_{\text{rel}} > 50 \vee v_{\text{rel}} \leq 50 \wedge p_{\text{rel}} - \frac{50^2}{2B} \geq 0$ underapproximates the atom for $v_{\text{rel}} \in [0, 100]$. We can thus append the following formula to our query: $\left(\text{accApprox} \to p_{\text{rel}} - \frac{v_{\text{rel}}^2}{2B} \geq 0\right)$. For $v_{\text{rel}} \in [0, 100]$ this formula is always satisfied[4].

---

[4]In practice, $v_{\text{rel}}$ may also be negative requiring a more complex approximation.

## B.2 Input Space Mosaics

The MOSAIC procedure takes a central role in the verification of nonlinear, non-normalized open-loop NNV queries. Classically, one uses DPLL(T) to decompose an arbitrary formula into conjunctions then handled by a theory solver. Open-loop NNV's crux is its use of reachability methods which do not lend themselves well to *classic* DPLL(T): Its usage would result in duplicate explorations of the same input space w.r.t. different output constraints which is inefficient. Therefore, we generalize DPLL(T) [40] through the MOSAIC procedure. The procedure receives a quantifier-free[5], non-normalized open-loop NNV query and enumerates *azulejos* of the input space each with an associated normalized linear open-loop NNV query $q_l$ (conjunction over input atoms, disjunctive normal form over output atoms) and nonlinear atoms in disjunctive normal form $q_n$. The input space is thus turned into a mosaic and the disjunction over all queries is *equivalent* to the input query. We can then obtain classical DPLL(T) by marking all atoms as linear input constraints. Our implementation of MOSAIC instruments a SAT solver on the Boolean skeleton of $p$ as well as a real arithmetic SMT solver to restructure a formula in this way.

For a formula $\zeta \in \text{FOL}_{\mathbb{R}}$, let sat-atoms$(\zeta)$ be the set of set of signed atoms such that for all $A \in \text{sat-atoms}(\zeta)$ it holds that $A$ only contains atoms of $\zeta$ or its negations ($A \subseteq \text{Atom}(\zeta) \cup \{\neg b \mid b \in \text{Atom}(\zeta)\}$). Further, we require for sat-atoms$(\zeta)$ that for any state $\nu$ it holds that $\nu \vDash \zeta$ iff there exists an $A \in \text{sat-atoms}(\zeta)$ such that $\nu \vDash \bigwedge_{a \in A} a$. Note that there may exist multiple such sets in which case we can choose an arbitrary one. For example, for $\zeta \equiv x > 0 \vee \neg (y > 0)$ we could get sat-atoms$(\zeta) = \{\{x > 0\}, \{\neg (x > 0), \neg (y > 0)\}\}$. For a given formula $\zeta$, we will call $J_\zeta$ the set of input variables. We introduce the following notation for projection of sat-atoms on the set $J_\zeta$:

$$\text{sat-atoms}(\zeta) \!\restriction_{J_\zeta} = \{\{a \mid a \in A \wedge \text{V}(a) \subseteq J_\zeta\} \mid A \in \text{sat-atoms}(\zeta)\}.$$

For example, reconsidering the previous example with $J_\zeta = \{x\}$ we would get sat-atoms$(\zeta)\!\restriction_{J_\zeta} = \{\{x > 0\}, \{\neg (x > 0)\}\}$. For a given open-loop NNV query $p_o$ and a given set of input variables $J_{p_o}$, MOSAIC then initially enumerates feasible combinations of linear input atoms (the azulejos) and for each such combination feasible combinations of mixed/output atoms are enumerated. This results in the following set:

$$S_1 = \left\{ \left( \bigwedge_{\substack{a \in i \\ a \in \text{FOL}_{\text{LR}}}} a \right) \wedge \left( \bigvee_{o \in \text{sat-atoms}(p_o \wedge i)} \left( \bigwedge_{\substack{b \in o \\ b \in \text{FOL}_{\text{LR}}}} b \right) \right) \;\middle|\; i \in \text{sat-atoms}(p_o)\!\restriction_{J_{p_o}} \right\}$$

Finally, for each $q_l \in S_1$, we can generate all possible combinations of nonlinear atoms $S_2 = \text{sat-atoms}(p_o \wedge q_l)$ and generate their disjunction:

$$q_n \equiv \bigvee_{A \in S_2} \bigwedge_{a \in A} a \tag{7}$$

We achieve this by enumerating all satisfying assignments for the boolean skeleton of $p_o$ (i.e. the formula where all atoms are substituted by boolean variables) using an incremental SAT solver. This initially happens in the same manner as it is done for the classical version of DPLL(T). However, once a model is found, we fix the assignment of linear input-only atoms and enumerate all other satisfying assignments of linear atoms, generating the disjunctions within $S_1$. For each conjunction we additionally enumerate possible assignments for the nonlinear atoms. Notably, through the encoding of LINEARIZE the procedure automatically knows which truth-combinations of a nonlinear constraint and its approximations may appear. We additionally provide information on linear dependencies between linear atoms to the SAT solver. All enumeration procedures are interleaved with calls to SMT solvers for linear and polynomial real arithmetic constraints, which check whether a given combination of constraints is indeed also satisfiable in the theory of real arithmetic (i.e. when interpreting the atoms as real arithmetic constraints instead of as boolean variables). In order to discard unsatisfiable solutions more quickly, we make use of unsatisfiability cores and conversely use a cache for satisfiable assignment combinations. We exploit partial models returned by the SAT solver to omit atoms which can (potentially) appear in both polarities for a given combination of constraints.

---

[5]A quantifier-free $p$ can be assumed as real arithmetic admits quantifier elimination [90]. In practice, all queries of interest were already quantifier-free.

We show that the decomposition is correct (i.e. the disjunction over all queries is equivalent to the original query, see Proposition 9) and that it is minimal in the sense that the resulting azulejos do not overlap (see Proposition 10) with proofs on pages 28 and 29:

**Proposition 9** (Correctness of Mosaic). *Let $p$ be any open-loop NNV query. Let $Q \subset FOL_{\mathbb{LR}} \times FOL_{\mathbb{R}}$ be the set returned by* MOSAIC($p$), *then the following formula is valid: $p \leftrightarrow \left( \bigvee_{(q_l, q_n) \in Q} (q_l \wedge q_n) \right)$*

**Proposition 10** (Flatness of Mosaic). *Let $\left( i_1 \wedge \bigvee_j o_{1,j} \right), \left( i_2 \wedge \bigvee_j o_{2,j} \right)$ be two linear queries enumerated by* MOSAIC *then $i_1 \wedge i_2$ is unsatisfiable.*

We could use this approach to decompose a nonlinear formula into a set of normalized linear open-loop NNV queries without approximation. MOSAIC then soundly omits all nonlinear constraints. However, this leads to many spurious counterexamples. Therefore, we add linear approximations (Appendix B.1) of atoms which are then automatically part of the conjunctions returned by MOSAIC.

**Running Example.**  In the previous section we extended our query by a linear underapproximation. Our procedure generates an azulejo for the case where $p_{\text{rel}} - \frac{100^2}{2B} \geq 0 \wedge v_{\text{rel}} > 50$ is satisfied (implying $p_{\text{rel}} - \frac{v_{\text{rel}}^2}{2B} \geq 0$) and the case where it is not. While the linear approximation is an edge of the mosaic tile, the original atom (the tile's "painting" describing the precise constraint) would be part of the *nonlinear* disjunctive normal form. For each azulejo, the output conjunctions of accCtrlFml are enumerated. For OVERT's approximation with $N = 1$, our implementation decomposes the ACC query into 20 normalized queries with up to 10 cases in the output constraint disjunction. Without MOSAIC each case would be treated as a separate reachability query leading to significant duplicate work.

**Relation to DPLL(T)**  Abstracting away the real-arithmetic, the MOSAIC algorithm generates tuples of normalized open-loop NNV queries and disjunctive normal forms that are satisfiable w.r.t. a theory solver $T$. The algorithm itself interleaves SAT-based reasoning about a boolean abstraction (annotated with information on whether an atom is linear and/or an input constraint) and theory solver invocations. We can now consider the case where all atoms (independent of their concrete contents and the theory $T$) are annotated as linear input constraints: In this case MOSAIC merely returns a mosaic of this "input" space where each azulejo corresponds to a conjunction of atoms that is satisfiable w.r.t. to the theory solver $T$ and the disjunction over all those conjunctions is then once again equivalent to the original formula – corresponding to DPLL(T)'s behavior.

### B.3  Counterexample Generalization and Enumeration

The innermost component of our algorithm enumerates all counterexample regions (ENUM). To this end, ENUM requires an algorithm which generalizes counterexample *points* returned by open-loop NNV to *regions* (GENERALIZE). For each such counterexample region we can then check if there exist concrete violations of the nonlinear constraints (FILTER). We begin by explaining GENERALIZE which converts a counterexample point returned by open-loop NNV into a counterexample region. The key insight for this approach is that a concrete counterexample $(z_0, x_0)$ returned by an open-loop NNV tool induces a region of points with *similar behavior* in the NN. A concrete input $z_0$ induces a fixed activation pattern for all piece-wise linear activations within the NN in a region $\iota$ around $z_0$. Consider the first layer's activation function $f^{(1)}$: $f^{(1)}$ can be decomposed into linear functions $f_i$ and so is a sum of affine transformations $A_i z_0 + b_i$ which are active iff $q_i(z_0)$ is true. We can then describe $f^{(1)}$'s local behavior around $z_0$ as the linear combination of all affine transformations active for $z_0$. This sum is itself an affine transformation. By iterating this approach across layers, we obtain a single affine transformation $\omega$ describing the NN's behavior in $\iota$. The regions returned by GENERALIZE are then defined as follows:

**Definition 11** (Counterexample Region). *For a given open-loop NNV query $q$ and piece-wise linear NN $g$, let $(z_0, x_0) \in \mathbb{R}^I \times \mathbb{R}^O$ be a counterexample, i.e. $x_0 = g(z_0)$ and $q(z_0, x_0)$ holds. The* counterexample region *for $z_0$ is the maximal polytope $\iota \subset \mathbb{R}^I$ with a linear function $\omega$ s.t. $z_0 \in \iota$ and $\omega(z) = g(z)$ for all $z \in \iota$.*

Star Sets [12, 93] can compute $(\iota, \omega)$ by steering the Star Set according to the activations of $z_0$. As the number of counterexample regions is exponentially bounded by the number of piece-wise linear nodes, we can use GENERALIZE for exhaustive enumeration. This is only a worst-case bound due to the NP-completeness of NN verification [58, 84]. In practice, the number of regions is much lower

since many activation functions are linear in all considered states. While a given counterexample region certainly has a point violating the *linear* query that was given to the open-loop NNV tool, it may be the case that the counterexample is *spurious*, i.e. it does not violate the nonlinear constraints. However, we can use the concise description of counterexample regions to check whether this is the case: The function $\omega$ describes the NN's *entire* behavior within $\iota$ as a single affine transformation and is thus much better suited for SMT-based reasoning. This SMT-based check is performed by FILTER based on the following insight:

**Lemma 12** (Counterexample Filter). *Let $(q_l, q_n)$ be a tuple returned by* MOSAIC. *A counterexample region $(\iota, \omega)$ for $q_l$ is a counterexample region for $q_l \wedge q_n$ iff the formula $\eta \equiv (q_l(z, x^+) \wedge q_n(z, x^+) \wedge z \in \iota \wedge x^+ = \omega(z))$ is satisfiable.*

See proof on page 29. The size of the formula $\eta$ only depends on $q_l$, $q_n$, $I$, and $O$ and, crucially, is *independent* of the size and architecture of the NN. In practice, even $x^+$ can be eliminated (substitute linear terms of $\omega(x^+)$).

Based on these insights, the last required component is a mechanism for the exhaustive enumeration of all counterexample regions (denoted as ENUM). There are two options for ENUM: Either we use geometric path enumeration [12, 93] to enumerate all counterexample regions (used for the evaluation) or we instrument arbitrary complete off-the-shelf open-loop NNV tools for linear queries through Algorithm 2. We define ENUM as follows:

**Definition 13** (Exhaustive Counterexample Generation). *An exhaustive enumeration procedure* ENUM *receives a linear, normalized open-loop NNV query $q$ and a piece-wise linear NN $g$ and returns a covering $E$ of counterexample regions, i.e. $E$ satisfies $\{z \in \mathbb{R}^I \mid q(z, g(z))\} \subseteq \bigcup_{(\iota, \omega) \in E} \iota$.*

---

**Algorithm 2:** Enumeration of counterexample regions using off-the-shelf open-loop NNV tools.

---

**Input:** Query $\bar{p}$, feed forward neural network (FNN) $g$
  **procedure** ENUMERATE($\bar{p}, g$)
    $s, E \leftarrow \mathtt{sat}, \varnothing$
    **while** $s = \mathtt{sat}$ **do**
      $s, e \leftarrow \mathrm{NNV}(\bar{p}, g)$                             $\triangleright$ Call open-loop NNV tool
      **if** $s = \mathtt{sat}$ **then**
        $\iota, \omega \leftarrow$ GENERALIZE$(e, g)$                   $\triangleright$ Generalize counterexample
        $E \leftarrow E \cup \{(\iota, \omega)\}$                       $\triangleright$ Store counterexample
        $\bar{p} \leftarrow \bar{p} \wedge \neg \iota$      $\triangleright$ Exclude counterexample region from remaining search space
    **return** $E$

---

## C   Proofs

### C.1   Proofs for Section 3

This subsection proves the soundness of the approach outlined in Section 3. This result is achieved by proving that a concrete NNCS refines [65, 79] an abstract hybrid program. The approach can either be applied by first proving safety for a suitable dL model or by reusing results from the dL literature (both demonstrated Section 5 and appendix E). In our proofs for Section 3 we show a slightly more general version of the result from Theorem 2 (see Theorem 21). To this end, we formally define a controller description as follows:

**Definition 14** (Controller Description). *Let $\alpha_{ctrl}$ be some hybrid program with free variables $FV(\alpha_{ctrl}) = \{z_1, \ldots, z_m\}$ and bound variables $BV(\alpha_{ctrl}) = \{x_1, \ldots, x_n\}$, which overlap if a variable is both read and written to. A controller description $\kappa \in FOL_{\mathrm{N\mathbb{R}}}$ for $\alpha_{ctrl}$ is a formula with free variables $FV(\alpha_{ctrl}) \cup \{x^+ \mid x \in BV(\alpha_{ctrl})\}$ such that the following formula is valid: $\forall z_1 \ldots z_m \exists x_1^+ \ldots x_n^+ \kappa$.*

Based on Controller Descriptions we can then show that such controller descriptions exist for all piece-wise Noetherian NNs:

**Lemma 15** (Existence of $\kappa_g$). *Let $g : \mathbb{R}^I \to \mathbb{R}^O$ be a piece-wise Noetherian NN. There exists a controller description $\kappa_g \in FOL_{\mathrm{N\mathbb{R}}}$ with input variables $z_1, \ldots, z_I$ and output variables $x_1^+, \ldots, x_O^+$ s.t.*

$\left(\nu(x_1^+), \ldots, \nu(x_O^+)\right) = g\left(\nu(z_1), \ldots, \nu(z_I)\right)$ *iff* $\nu \vDash \kappa_g$, *i.e.* $\kappa_g$*'s satisfying assignments correspond exactly to g's in-out relation.*

*Proof.* Previous work showed how to encode piece-wise linear NNs through real arithmetic SMT formulas (see e.g. [36, 72]). Each output dimension of an affine transformation can be directly encoded as a real arithmetic term. For a given output-dimension of a piece-wise Noetherian activation function we have to encode a term $\sum_{i=1}^{s} \mathbb{1}_{q_i}(x) f_i(x)$ with Noetherian functions $f_i$ and predicates $q_i$ over real arithmetic with Noetherian functions. To this end, we can introduce fresh variables $v_1, \ldots, v_s$ where we assert the following formula for each $v_i$:

$$\left(q_i \wedge v_i = f_i(x)\right) \vee \left(\neg q_i \wedge v_i = 0\right).$$

The activation function's result then is the sum $\sum_{i=1}^{s} v_i$. By existentially quantifying all intermediate variables of such encodings, we obtain a real arithmetic formula that only contains input and output variables and satisfies the requirements of the Lemma.

If we assign $x_1^+ \ldots x_O^+$ to the values provided by $g(z_1, \ldots, z_I)$ for a given $z_1 \ldots z_I$, the formula $\kappa_g$ is satisfied. Therefore, $\forall z_1 \ldots z_I \exists x_1^+ \ldots x_O^+ \kappa_g$. is valid. □

When replacing $\alpha_{ctrl}$ by an NN $g$, free and bound variables of $\alpha_{ctrl}$ must resp. match to input and output variables of $g$. Based of a description $\kappa_g$, we then construct a hybrid program that behaves as described by $\kappa_g$: We now formalize the idea of modeling a given NN $g$ through a hybrid program which behaves identically to $g$.We show that such *nondeterministic mirrors exist* for all piece-wise Noetherian NN $g$:

**Definition 16** (Nondeterministic Mirror for $\kappa_g$). *Let $\alpha_{ctrl}$ be some hybrid program with bound variables $BV(\alpha_{ctrl}) = \{x_1, \ldots, x_n\}$. For a controller description $\kappa_g$ with variables matching to $\alpha_{ctrl}$, $\kappa_g$'s nondeterministic mirror $\alpha_{refl}$ is defined as:*
$\alpha_{refl}(\kappa_g) \equiv (x_1^+ := *; \ldots; x_n^+ := *; ?(\kappa_g); x_1 := x_1^+; \ldots; x_n := x_n^+)$

**Lemma 17** (Existence of $\alpha_g$). *For any piece-wise Noetherian NN $g : \mathbb{R}^I \to \mathbb{R}^O$ there exists a nondeterministic mirror $\alpha_g$ that behaves identically to g.*
*Formally, $\alpha_g$ only has free variables $\overline{z}$ and bound variables $\overline{x}$ and for any state transition $(\nu, \mu) \in [\![\alpha_g]\!]$: $\mu(\overline{x}) = g(\nu(\overline{z}))$ ($\overline{x}$ and $\overline{z}$ vectors of dimension I and O)*

*Proof.* Based on Lemma 15 we can construct a controller description $\kappa_g \in \text{FOL}_{\text{N}\mathbb{R}}$ for $g$ which we can turn into a hybrid program through the nondeterministic mirror $\alpha_{refl}(\kappa_g)$. □

Similarly to the more general notion of a controller description, Theorem 21 also permits a slightly more general version of a state space restriction instead of an inductive invariant. Formally, this notion is described as a state reachability formula:

**Definition 18** (State Reachability Formula). *A state reachability formula $\zeta_s$ with free variables $z_1, \ldots, z_m$ is* complete *for the hybrid program $(\alpha_{ctrl}; \alpha_{plant})^*$ with free variables $z_1, \ldots, z_m$ and initial state $\phi$ iff the following* dL *formula is valid where $(\zeta_s)_{z_1 \ldots z_m}^{z_1^+ \ldots z_m^+}$ represents $\zeta_s$ with $z_i^+$ substituted for $z_i$ for all $1 \leq i \leq m$:*

$$\left(\phi \wedge \langle(\alpha_{ctl}; \alpha_{plant})^*\rangle \bigwedge_{i=1}^{m} z_i = z_i^+\right) \to (\zeta_s)_{z_1 \ldots z_m}^{z_1^+ \ldots z_m^+}. \tag{8}$$

There is usually an overlap between free and bound variables, i.e. $z_1, \ldots, z_m$ may contain variables later modified by the hybrid program. Our definition requires that for any program starting in a state satisfying $\phi$, formula $\zeta_s$ is satisfied in all terminating states. $\zeta_s$ thus overapproximates the program's reachable states. In particular, inductive invariants (i.e. for $\phi \to [\alpha^*]\psi$ a formula $\zeta$ s.t. $\phi \to \zeta$ and $\zeta \to [\alpha]\zeta$) are state reachability formulas:

**Lemma 19** (Inductive Invariants are State Reachability Formulas). *If $\zeta$ is an inductive invariant of $\phi \to [(\alpha_{ctrl}; \alpha_{plant})^*]\psi$, $\zeta$ is a state reachability formula.*

*Proof.* We begin by recalling the requirement for $\zeta$ to be a state reachability formula:

$$\left(\phi \wedge \langle(\alpha_{ctl}; \alpha_{plant})^*\rangle \bigwedge_{i=1}^{m} z_i = z_i^+\right) \to (\zeta_s)_{z_1 \ldots z_m}^{z_1^+ \ldots z_m^+}.$$

Let $\zeta$ be an inductive invariant for some contract of the form given above. First, consider that for any state satisfying the left side of the formula above it holds that there exists some $k$ such that $\phi \wedge \langle \left( \alpha_{\text{ctrl}}; \alpha_{\text{plant}} \right)^k \rangle \bigwedge_{i=1}^m z_i = z_i^+$ is satisfied by the same state (this follows from the semantics of loops in dL). If we can prove that any such state also satisfies $(\zeta_s)_{z_1,\dots,z_m}^{z_1^+,\dots,z_m^+}$ we obtain that $\zeta$ is a state reachability formula. We proceed by induction: First, consider $k = 0$ in this case $z_i$ has the same value as $z_i^+$ for all $i$. The formula then boils down to $\phi \to \zeta$. This formula is guaranteed to be valid by the first requirement of inductive invariants. Next, we now assume that we already proved that $\zeta$ holds for $k$ loop iterations and show it for $k + 1$. Since we assume some state $u_{k+1}$ that satisfies $\phi \wedge \langle \left( \alpha_{\text{ctrl}}; \alpha_{\text{plant}} \right)^{k+1} \rangle \bigwedge_{i=1}^m z_i = z_i^+$, there also has to be some state $u_k$ satisfying $\phi \wedge \langle \left( \alpha_{\text{ctrl}}; \alpha_{\text{plant}} \right)^k \rangle \bigwedge_{i=1}^m z_i = z_i^+$. However, we already know for $u_k$ that it satisfies $(\zeta_s)_{z_1,\dots,z_m}^{z_1^+,\dots,z_m^+}$. Since $u_{k+1}$ is reachable from $u_k$ through the execution of $\alpha_{\text{ctrl}}; \alpha_{\text{plant}}$ we know that $u_{k+1}$ satisfies $(\zeta_s)_{z_1,\dots,z_m}^{z_1^+,\dots,z_m^+}$ (this corresponds to the property $\zeta \to [\alpha]\zeta$ of inductive invariants). $\qquad\square$

As $\alpha_g$ and $g$ mirror each other, we can reason about them interchangeably. Our objective is now to prove that $\alpha_g$ is a refinement of $\alpha_{\text{ctrl}}$. To this end, we use the shielding technique ModelPlex [69] to automatically generate a correct-by-construction controller monitor $\zeta_c$ for $\alpha_{\text{ctrl}}$. The formula $\zeta_c$ then describes what behavior for $\alpha_g$ is acceptable so that $\alpha_g$ still represents a refinement of $\alpha_{\text{ctrl}}$. As seen in Section 3, we do *not* require that $\alpha_g$ adheres to $\zeta_c$ on all states, but only on reachable states. For efficiency we therefore allow limiting the analyzed state space to an inductive invariant $\zeta_s$ (i.e. for $\phi \to [\alpha^*]\psi$ a formula $\zeta_s$ s.t. $\phi \to \zeta_s$ and $\zeta_s \to [\alpha]\zeta_s$). Despite the infinite-time horizon, the practical use of our approach often faces implementations with a limited value range for inputs and outputs (e.g., $v_{\text{rel}}$ within the ego-car's physical capabilities). Only by exploiting these ranges, is it possible to prove safety for NNs that were only trained on a particular value range. To this end, we allow specifying value *ranges* (i.e. intervals) for variables. We define the range formula $R \equiv \bigwedge_{v \in V(P)} \underline{R}(v) \le v \le \overline{R}(v)$ for lower and upper bounds $\underline{R}$ and $\overline{R}$. Using $R$, we specialize a contract to the implementation specifics by adding a range check to $\alpha_{\text{plant}}$. The safety results for the original contract can be reused:

**Lemma 20** (Range Restriction). *Let $\phi \to \left[ \left( \alpha_{ctl}; \alpha_{plant} \right)^* \right] \psi$ be a valid dL formula. Then the formula $C_2 \equiv \left( \phi \wedge R \to \left[ \left( \alpha_{ctrl}; \left( \alpha_{plant}; ?(R) \right) \right)^* \right] \psi \right)$ with ranges $R$ is valid and $R$ is an invariant for $C_2$.*

*Proof.* We use the notation $C_1 \equiv \left( \phi \to \left[ \left( \alpha_{\text{ctl}}; \alpha_{\text{plant}} \right)^* \right] \psi \right)$. We begin by showing that the validity of $C_1$ implies the validity of $C_2$. Intuitively, this follows from the fact that the introduced check $?(R)$ only takes away states. Let $I$ be a loop invariant such that $\phi \to I$, $I \to \psi$ and $I \to \left[ \alpha_{\text{ctrl}}; \alpha_{\text{plant}} \right] I$ (assumed due to the validity of $C_1$). Then clearly, it also holds that $\phi \wedge R \to I$. Furthermore, $I \to \left[ \alpha_{\text{ctrl}}; \alpha_{\text{plant}}; ?(R) \right] I$ can be reduced to $I \to \left[ \alpha_{\text{ctrl}}; \alpha_{\text{plant}} \right] (R \to I)$ which we can shown through the monotonicity rule of dL. Since we already know that $I \to \psi$, it follows that $C_2$ is valid, because $I$ is a loop invariant. $\qquad\square$

Including $R$ into $\zeta_s$ allows us to exploit the range limits for the analysis of $\alpha_g$. Our objective is to use open-loop NNV techniques to check whether $g$ (and therefore $\alpha_g$) satisfies the specification synthesized by ModelPlex. To this end, we use a *nonlinear NN verifier* to prove safety of our NNCS:

**Theorem 21** (Safety Criterion). *Let $\zeta_c$ and $\zeta_s$ be controller and state reachability formulas for a valid dL contract $C \equiv \left( \phi \to \left[ \left( \alpha_{ctl}; \alpha_{plant} \right)^* \right] \psi \right)$. For any controller description $\kappa$, if*

$$\zeta_s \wedge \kappa \to \zeta_c \tag{9}$$

*is valid, then the following dL formula is valid as well:*

$$\phi \to \left[ \left( \alpha_{refl}(\kappa); \alpha_{plant} \right)^* \right] \psi \tag{10}$$

*Proof.* Assume the validity of

$$\zeta_s \wedge \kappa \to \zeta_c.$$

Let $v \in \mathcal{S}$ be some arbitrary state. We need to show that any such $v$ satisfies Formula (10):

$$\phi \to \left[ \left( \alpha_{\text{refl}}(\kappa); \alpha_{\text{plant}} \right)^* \right] \psi.$$

To this end, assume $v \vDash \phi$, we prove that $\psi$ as well as $\zeta_s$ is upheld after any number of loop iterations by induction on the number $n$ of loop iterations.

*Base Case:* $n = 0$

In this case, the only state we need to consider is $v$ since there were no loop iterations. We know through the validity of $C$ that $\phi \to \psi$. Thus, $v \vDash \psi$. Furthermore, we recall the requirement of a state reachability formula:

$$\left( \phi \wedge \langle \left( \alpha_{\mathrm{ctl}}; \alpha_{\mathrm{plant}} \right)^* \rangle \bigwedge_{i=1}^{m} z_i = z_i^+ \right) \to (\zeta_s)_{z_1 \ldots z_m}^{z_1^+ \ldots z_m^+} .$$

By extending $v$ such that all $z_i^+$ have the same values as the corresponding $z_i$, we get a state that satisfies this formula. Consequently, $v \vDash \zeta_s$.

*Inductive Case:* $n \to (n+1)$

In the induction case, we know that for all

$$(v, \tilde{v}_0) \in \left[\!\left[ \left( \alpha_{\mathrm{refl}}(\kappa); \alpha_{\mathrm{plant}} \right)^n \right]\!\right]$$

it holds that $\tilde{v}_0 \vDash \psi$ and $\tilde{v}_0 \vDash \zeta_s$.

We must now prove the induction property for any state reachable from $\tilde{v}_0$ through execution of the program $\left( \alpha_{\mathrm{refl}}(\kappa); \alpha_{\mathrm{plant}} \right)$. For any $\tilde{v}_1 \in \mathcal{S}$ such that $(\tilde{v}_0, \tilde{v}_1) \in \left[\!\left[ x_1^+ := *; \ldots; x_n^+ := *; ?(\kappa); \right]\!\right]$ (by the definition of $\kappa$ we know that such a state exists) we know that $\tilde{v}_1 \vDash \kappa$. According to the coincidence lemma [69, Lemma 3], since $\zeta_s$ does not concern the $x^+$ variables, it is then true that

$$\tilde{v}_1 \vDash \zeta_s \wedge \kappa.$$

Through the validity of Formula (9) assumed at the beginning, we then know that it must be the case that $\tilde{v}_1 \vDash \zeta_c$. More specifically, this means that for any $\tilde{v}_2 \in \mathcal{S}$ with

$$(\tilde{v}_1, \tilde{v}_2) \in \left[\!\left[ x_1 := x_1^+; \ldots; x_n := x_n^+ \right]\!\right];$$

it holds that $(\tilde{v}_0, \tilde{v}_2) \vDash \zeta_c$. By definition this implies that $(\tilde{v}_0, \tilde{v}_2) \in \left[\!\left[ \alpha_{\mathrm{ctrl}} \right]\!\right]$.

In summary, this means that for any $(\tilde{v}_0, \tilde{v}_2) \in \left[\!\left[ \alpha_{\mathrm{refl}}(\kappa) \right]\!\right]$ it holds that $(\tilde{v}_0, \tilde{v}_2) \in \left[\!\left[ \alpha_{\mathrm{ctrl}} \right]\!\right]$.

Through the semantics of program composition in hybrid programs it follows that subsequently for any $\tilde{v}_3 \in \mathcal{S}$ with $(\tilde{v}_0, \tilde{v}_3) \in \left[\!\left[ \alpha_{\mathrm{refl}}(\kappa); \alpha_{\mathrm{plant}} \right]\!\right]$ it holds that $(\tilde{v}_0, \tilde{v}_3) \in \left[\!\left[ \alpha_{\mathrm{ctrl}}; \alpha_{\mathrm{plant}} \right]\!\right]$.

We also know that $(v, \tilde{v}_0) \in \left[\!\left[ \left( \alpha_{\mathrm{ctl}}; \alpha_{\mathrm{plant}} \right)^* \right]\!\right]$ and that $(\tilde{v}_0, \tilde{v}_3) \in \left[\!\left[ \alpha_{\mathrm{ctrl}}; \alpha_{\mathrm{plant}} \right]\!\right]$. Since this implies $(v, \tilde{v}_3) \in \left[\!\left[ \left( \alpha_{\mathrm{ctl}}; \alpha_{\mathrm{plant}} \right)^* \right]\!\right]$, i.e. there is a trace of states from $v$ to $\tilde{v}_3$, and since we already know that $v \vDash \phi$, $(v, \tilde{v}_3)$ satisfy the right side of Formula (8). Since Formula (8) must be valid we get that $\tilde{v}_3 \vDash \zeta_s$. Consequently, we know through the validity of $C$ that:

$$\tilde{v}_3 \vDash \psi \wedge \zeta_s.$$

This concludes the induction proof and thereby also the proof of Theorem 21 $\qquad\square$

**Definition 22** (Nonlinear Neural Network Verifier). *A nonlinear neural network verifier* accepts *as input a piece-wise Noetherian NN $g$ and nonlinear open-loop NNV query $p$ with free variables $z_1, \ldots, z_I, x_1^+, \ldots, x_O^+$. The tool must be* sound, *i.e. if there is a $z \in \mathbb{R}^I$ satisfying $p(x, g(z))$ then the tool must return* sat. *A tool that always returns* unsat *if no such $z \in \mathbb{R}^I$ exists is called* complete.

**Lemma 23** (Soundness w.r.t Controller Descriptions). *Let $\kappa_g$ be a controller description for a piece-wise Noetherian NN $g$.*
*Further, let $C \equiv \left( \phi \to \left[ \left( \alpha_{ctl}; \alpha_{plant} \right)^* \right] \psi \right)$ be a contract with controller monitor $\zeta_c \in FOL_{\mathbb{R}}$ and inductive invariant $\zeta_s \in FOL_{\mathbb{R}}$ where the free and bound variables respectively match $g$'s inputs and outputs. If a sound Nonlinear Neural Network Verifier returns* unsat *for the query $p \equiv (\zeta_s \wedge \neg \zeta_c)$ on $g$ then: 1. $\kappa_g \wedge \zeta_s \to \zeta_c$ is valid; 2. $\phi \to \left[ \left( \alpha_{refl}(\kappa_g); \alpha_{plant} \right)^* \right] \psi$ is valid.*

*Proof.* Let all variables be defined as above. We assume that the nonlinear NN verifier did indeed return `unsat`. By definition this means that there exists no $z \in \mathbb{R}^I$ such that $p(z, g(z)) = \top$. Due to the formalization of $\kappa_g$ (see Lemma 15), this means there exists no $z \in \mathbb{R}^I$ such that $\zeta_s \wedge \kappa_g \wedge \neg\zeta_c$. Among all states consider now any state $v$ such that $v \not\models \zeta_s \wedge \kappa_g$. In this case $v \models \zeta_s \wedge \kappa_g \rightarrow \zeta_c$ vacuously. Next, consider the other case, i.e. a state $v$ such that $v \models \zeta_s \wedge \kappa_g$. In this case it must hold that $v \not\models \neg\zeta_c$. So $v \models \zeta_c$. Therefore, $v \models \zeta_s \wedge \kappa_g \rightarrow \zeta_c$. This means that Formula (9) is satisfied by all states and, therefore, valid which proves the first claim. Theorem 21 then implies the safety guarantee stated in Formula (10) for $\kappa_g$ which proves the second claim. □

While we lay the foundation for analyses on piece-wise Noetherian NNs, a subclass is decidable:

**Lemma 24** (Decidability for Polynomial Constraints). *Given a piece-wise polynomial NN $g$, the problem of verifying $(\zeta_s \wedge \neg\zeta_c) \in FOL_\mathbb{R}$ for $g$ is decidable.*

*Proof.* The problem of verifying $(\zeta_s \wedge \neg\zeta_c) \in \mathrm{FOL}_\mathbb{R}$ for $g$ is the same as proving the validity of the formula $\zeta_s \wedge \kappa_g \rightarrow \zeta_c$ (see Lemma 23). For piece-wise polynomial NN this formula is in $\mathrm{FOL}_\mathbb{R}$ and the validity problem is thus decidable. □

*Proof of 2.* The formula $\phi \rightarrow \left[\left(\alpha_g ; \alpha_{\mathrm{plant}}\right)^*\right]\psi$ is equivalent to $\phi \rightarrow \left[\left(\alpha_{\mathrm{refl}}\left(\kappa_g\right) ; \alpha_{\mathrm{plant}}\right)^*\right]\psi$ as $\alpha_g$ behaves precisely like $\alpha_{\mathrm{refl}}\left(\kappa_g\right)$. The result immediately follows from Lemma 23 □

## C.2 Proofs for Section 4

*Proof of 3.* This proof assumes the results from Appendices B.1 to B.3. We begin by proving soundness, i.e. if Algorithm 1 returns safe, then there exists no counterexample. Consider a counterexample region found by ENUM. Lemma 12 tells us that this counterexample can only be concrete if formula $\nu$ is satisfied. This is the check performed by FILTER. Thus, Algorithm 1 only skips a counterexample region if it is not concrete. Further, we know through Definition 13 that all counterexamples are returned by the procedure for a given query $q_l$. Further, we know that the disjunction over all $q_l \wedge q_n$ returned by MOSAIC is equivalent to its input $p_o$ (Proposition 9) and thus the disjunction over all $q_l$ is an over-approximation thereof. Finally, LINEARIZE returns a formula $p_o$ which is equivalent to the input query $p$ (Lemma 6). Therefore, any counterexample of $p$ must also be a counterexample of some $q_l$ returned by MOSAIC. Consequently, we iterate over all possible counterexamples and only discard them if they are spurious. Thus, our algorithm is sound.

We now turn to the question of completeness, i.e. we prove that any time Algorithm 1 returns `unsafe` then there is indeed a concrete counterexample of $p$. First, remember that Lemma 6 ensures that $p$ and $p_o$ are equivalent. Furthermore, Proposition 9 ensures that the disjunction over all $q_l \wedge q_n$ generated by MOSAIC is equivalent to $p_o$. Assume we found a counterexample. The algorithm will return `unsafe` iff FILTER returns that the counterexample is concrete. According to Lemma 12 we know that this is only the case if there is indeed a concrete counterexample for $q_l \wedge q_n$. Since this counterexample then also satisfies $p_o$ (see above), we only return `unsafe` if FILTER found a concrete counterexample for $p$. As real arithmetic is decidable and all other procedures in the algorithm terminate as well, this yields a terminating, sound and complete algorithm. □

*Proof of 6.* Let $p \in \mathrm{FOL}_\mathbb{R}$ be some nonlinear open-loop NNV query and $a_1, \ldots, a_k$ be the nonlinear atoms in $p$. From the definition of LINEARIZE, we know that $p_o$ has the form $p \wedge \bigwedge_{i=1}^k \left(\left(a_i \rightarrow \overline{a_i}\right) \wedge \left(\underline{a_i} \rightarrow a_i\right)\right)$. By definition, it holds for approximations $\underline{a_i}, \overline{a_i}$ that for any state $\nu$ with $\nu \models a_i \wedge R$ it also holds that $\nu \models \overline{a_i}$ (resp. for any state $\nu$ with $\nu \models \underline{a_i} \wedge R$ it also holds that $\nu \models a_i$). Consequently, the formulas $R \rightarrow (a_i \rightarrow \overline{a_i})$ and $R \rightarrow \left(\underline{a_i} \rightarrow a_i\right)$ are valid for all $a_i$. Let $\nu$ be a state such that $\nu \models p$. Then, by definition $\nu \models R$ and due to the above mentioned validity it therefore holds that $\nu \models a_i \rightarrow \overline{a_i}$ and $\nu \models \underline{a_i} \rightarrow a_i$. Therefore, $\nu \models p_o$. Conversely, for any state with $\nu \models p_o$ it also holds that $\nu \models p$. □

*Proof of 9.* We begin by considering the case where some state $\nu$ satisfies $\bigvee_{(q_l, q_n) \in Q} q_l \wedge q_n$. By definition, this means that there exists some $(q_l^*, q_n^*) \in Q$ such that $\nu \models q_l \wedge q_n$. Through the definition of the set $S_1$ in Appendix B, we know that $q_l^*$ contains a conjunction over linear input atoms $i_l^*$. Let $o_l^* \in \text{sat-atoms}(p \wedge i_l^*)$ be the set of mixed/output atoms such that $\nu \models \bigwedge_{b \in o^*} b$. Further, since $\nu \models q_n$,

we know there also exists an $A^* \in$ sat-atoms$(p \wedge q_l)$ such that $\nu \vDash \bigwedge_{a^* \in A^*} a^*$. Through the definition of sat-atoms and its projection we then know that $A^* \cup i^* \cup o^* \in$ sat-atoms$(p)$. Consequently, it must hold that $\nu \vDash p$.

Consider now the other direction where for some state $\nu$ it holds that $\nu \vDash p$. By definition of sat-atoms, its projection and $S_1$ we know that there must exist some $i^* \in$ sat-atoms$(p)|_{J_p}$ such that $\nu \vDash i^*$. Moreover, there must exist an $o \in$ sat-atoms$(p \wedge i^*)$ such that $\nu \vDash \bigwedge_{b \in o^*} b$. Finally, since $\nu \vDash i^* \wedge o$, there must exist an $A^* \in$ sat-atoms$(p \wedge i^* \wedge o)$ such that $\nu \vDash \bigwedge_{a^* \in A^*} a^*$ Consequently, there exists a $(q_l, q_n) \in Q$ such that $\nu \vDash q_l \wedge q_n$ and therefore $\nu \vDash \bigvee_{(q_l,q_n) \in Q} q_l \wedge q_n$. $\qquad\square$

*Proof of 10.* Assume there were two linear queries $\left(i_1 \wedge \bigvee_j o_{1,j}\right)$ and $\left(i_2 \wedge \bigvee_j o_{2,j}\right)$ such that $i_1 \wedge i_2$ had a model. By definition, each set $A \in$ sat-atoms$(p)$ must contain each atom of $p$ or its negation. Consider now the projection sat-atoms$(p)|_{J_p}$ from which we obtain all $i$s (in particular $i_1$ and $i_2$): Since $i_1$ and $i_2$ contain the same set of atoms, it must be the case that for some atom $a \in i_1$, it holds that $\neg a \in i_2$ or vice versa (otherwise, the two would be identical). Through the law of the excluded middle, we get that $a \wedge \neg a$ is unsatisfiable, and thus $i_1 \wedge i_2$ is unsatisfiable. $\qquad\square$

*Proof of 12.* Assume some $(\iota, \omega)$ is indeed a counterexample region for $q_l \wedge q_n$. In this case, we know that there is some $z \in \iota$ such that with $x^+ = g(z)$ we get $q_l(z, x^+) \wedge q_n(z, x^+)$. However, by definition of counterexample regions we also know that $g(z) = \omega(z)$. Therefore, the assignments of $z$ and $x^+$ satisfy $\eta$. Next, consider the other direction. I.e. we assume we have a satisfying assignment for $\eta$. By definition we know that for the given assignment of $z$ it holds that $x^+ = g(z) = \omega(z)$. Therefore, $z, x^+$ respect the neural network and satisfy $q_l \wedge q_n$, which are the two requirements for a counterexample. $\qquad\square$

## D    Adaptive Cruise Control

**Information on the dL model.**    The controller $\alpha_{\text{ctrl}}$ has three nondeterministic options: it can brake with $-B$ (no constraints), set relative acceleration to $a_{\text{rel}} = 0$ (constraint accCtrl$_0$) or choose any value in the range $[-B, A]$ (constraint accCtrl$_1$). The constraints for the second and third action are as follows:

$$\text{accCtrl}_0 \equiv \left(2B\left(p_{\text{rel}} + Tv_{\text{rel}}\right) > v_{\text{rel}}^2\right)$$
$$\text{accCtrl}_1 \equiv 2B\left(p_{\text{rel}} + Tv_{\text{rel}} + 0.5T^2 a_{\text{rel}}\right) > \left(v_{\text{rel}} + Ta_{\text{rel}}\right)^2 \wedge$$
$$\left(-v_{\text{rel}} > Ta_{\text{rel}} \vee 0 < v_{\text{rel}} \vee \left(v_{\text{rel}}^2 < 2a_{\text{rel}}p_{\text{rel}}\right)\right)$$

We can prove the safety of this control envelope for the following initial condition which is also the loop invariant:

$$\text{accInit} \equiv \text{accInv} \equiv p_{\text{rel}} > 0 \wedge p_{\text{rel}} 2B \geq v_{\text{rel}}^2$$

The right-hand side of the invariant/initial condition ensures that the distance is still large enough to avoid a collision through an emergency brake ($a_{\text{rel}} = -B$). Based on these foundations, the full specification for the NN generated by VerSAILLE reads as follows:

$$\left(0 \leq p_{\text{rel}} \wedge p_{\text{rel}} \leq 100 \wedge -200 \leq v_{\text{rel}} \wedge v_{\text{rel}} \leq 200 \wedge\right.$$
$$-B \leq a_{\text{rel}}^+ \wedge a_{\text{rel}}^+ \leq A \wedge$$
$$\left.p_{\text{rel}} > 0 \wedge p_{\text{rel}} \geq v_{\text{rel}}^2/(2*B)\right) \rightarrow$$
$$\left(a_{\text{rel}}^+ \geq A = v_{\text{rel}} \vee\right.$$
$$a_{\text{rel}}^+ \geq -B \wedge a_{\text{rel}}^+ < A \wedge a_{\text{rel}}^+ \neq 0 \wedge$$
$$\left(\left(-v_{\text{rel}}/a_{\text{rel}}^+ > T \vee -v_{\text{rel}} < 0\right) \wedge\right.$$
$$p_{\text{rel}} + v_{\text{rel}} * T + a_{\text{rel}}^+ * T^2/2 > (v_{\text{rel}} + a_{\text{rel}}^+ * T)^2/(2*B) \vee$$
$$p_{\text{rel}} + v_{\text{rel}} * T + a_{\text{rel}}^+ * T^2/2 > (v_{\text{rel}} + a_{\text{rel}}^+ * T)^2/(2*B) \wedge$$
$$\left.p_{\text{rel}} * a_{\text{rel}}^+ - v_{\text{rel}}^2 + v_{\text{rel}}^2/2 > 0\right) \vee$$
$$\left.p_{\text{rel}} + v_{\text{rel}} * T > v_{\text{rel}}^2/(2*B) \wedge a_{\text{rel}}^+ = 0\right)$$

For our verification, we set $T = 0.1$ (note that this is a bound on the frequency of control decisions, not a time horizon) and $A = B = 100$.

# E    Extended Evaluation

We implemented our procedure in a new tool[6] called $N^3V$. Due to the widespread use of ReLU NNs, $N^3V$ focuses on the verification of generic open-loop NNV queries for such NNs, but could be extended in future work. Our tool is implemented in Julia [16] using nnenum [10, 12] for open-loop NNV, PicoSAT [17, 19] and Z3 [32, 52]. Our evaluation aimed at answering the following questions:

Q1   Can $N^3V$ verify infinite-time horizon safety or exhaustively enumerate counterexample regions for a given NNCS?

Q2   Does our approach advance the State-of-the-Art?

Q3   Does our approach scale to complex real-world scenarios such as ACAS X?

The case studies comprised continuous and discrete control outputs. (Q3) is answered in the paper's main evaluation (see Section 5; the remaining questions are discussed below. Times are wall-clock times on a 16 core AMD Ryzen 7 PRO 5850U CPU ($N^3V$ itself is sequential while nnenum uses multithreading).

## E.1    Verification of Adaptive Cruise Control

We applied our approach to the previously outlined running example. To this end, we trained two NNs using PPO [82]: `ACC` contains 2 layers with 64 ReLU nodes each while `ACC_Large` contains 4 layers with 64 ReLU nodes each. Our approach only analyzes the hybrid system *once* and reuses the formulas for all future verification tasks (e.g. after retraining). We analyzed both NNs and a third one (see below for details) using $N^3V$ for coarser and tighter approximation settings (using OVERT's setting $N \in \{1, 2, 3\}$) on the value range $(p_{rel}, v_{rel}) \in [0, 100] \times [-200, 200]$. The analyses took 47 to 300 seconds depending on the NN and approximation. The runtimes show mixed results for tighter approximations: While tighter approximations (i.e. a higher $N$) sometimes improves performance (e.g. for `ACC_Large` retrained), it can also harm performance (e.g. as seen for `ACC_Large`). We suspect that this is a combination of two factors. First, finer approximations yield a larger number of queries which may increase the overall overhead. Secondly, our adjustments to OVERT's approximation using approximation of piece-wise linearities (see Appendix B.1) may in some cases worsen the approximation in comparison to a lower $N$. We leave a more fine-grained analysis of approximation techniques to future work and focus our analysis on approximations with $N = 1$. Across all NNs we find that approximation helps: The first row shows performance when omitting the approximated constraints in the open-loop NNV query and uniformly performs worse than an $N = 1$ approximation. $N^3V$ finds the NN `ACC_Large` to be *unsafe* and provides an exhaustive characterization of all input space regions with unsafe actions.

**Information on the counterexamples found for `ACC_Large`.**    Figure 6 shows the input state where the x-axis represents possible values for $p_{rel}$ and the y-axis represents possible values for $v_{rel}$. The orange line represents the edge of the safe state space, i.e. all values below the orange line are outside the reachable state space of the contract. The red areas represent all parts of the state space where $N^3V$ found concrete counterexamples for the checked controller monitor formula. Furthermore, the plot contains two lines representing the system's evolution over time when started at certain initial states. In particular, we observe one trajectory leading to a crash due to an erroneous decision in the red area around $p_{rel} = 5, v_{rel} = -25$. This concrete counterexample was found by sampling initial states from the regions provided by $N^3V$.

**Information on runtimes.**    The runtimes can be seen in Table 5 where #Filtered corresponds to the number of counterexample regions that were found to be spurious for `ACC_Large`. Comparing linear and $N = 1$ performance we see that approximation especially helps for larger NNs (`ACC` vs `ACC_Large`). This is the case because the overapproximate constraints can filter out numerous counterexample regions, which would otherwise have to be processed by the FILTER procedure. This effect is less significant for smaller networks where the time for overapproximation construction

---

[6]See https://github.com/samysweb/NCubeV or [91]

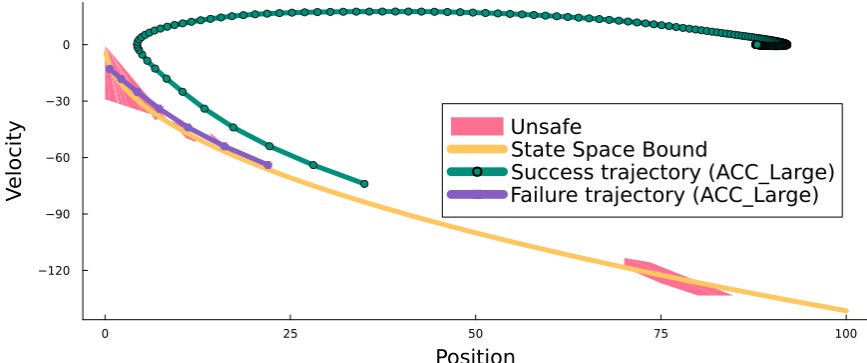

Figure 6: This plot shows the (input) state space for the `ACC_Large` NNCS: The orange line represents the boundary of the safe state space; the red areas indicate regions with counterexamples; the purple and green lines show potential trajectories of the system (dots represent discrete controller decisions).

| Approx. | ACC | ACC_Large | ACC_Large retrained | #Filtered |
|---------|-----|-----------|---------------------|-----------|
| linear  | 65s | 228s      | 277s                | 5658      |
| $N = 1$ | 47s | 87s       | 124s                | 1889      |
| $N = 2$ | 53s | 300s      | 70s                 | 5604      |
| $N = 3$ | 92s | 139s      | 102s                | 1306      |

Table 5: Runtime of $N^3V$ on the ACC networks per approximation. Final column lists filtered counterexamples for `ACC_Large`

takes longer in comparison to the NN analysis time and less counterexamples are generated due to the lower number of ReLU nodes.

**Further Training.** Approx. 3% of `ACC_Large`'s inputs resulted in unsafe actions which demonstrably resulted in car crashes. We performed a second training round on `ACC_Large` where we initialized the system within the counterexample regions for a boosted $p \approx 13\%$ of all runs (choosing the best-performing $p$). By iterating this approach twice, we obtained an NN which was safe except for very small relative distances (for $(p_{rel}, v_{rel}) \in [0, 0.08] \times [-2, 0.1]$). $N^3V$ certifies the safety outside this remaining region (see column `ACC_Large` retrained) which can be safeguarded using an emergency braking backup controller. Notably, this an *a priori* guarantee is for an arbitrarily long trip.

**Results on Q1.** Our tool $N^3V$ is capable of verifying and refuting infinite-time horizon safety for a given dL contract. The support for exhaustively enumerating counterexamples can help in guiding the development of safer NNCSs.

### E.2 Comparison to Other Techniques

Although Closed-loop NNV tools focus on finite-time horizons, we did compare our approach with the tools from ARCH Comp 2022 [66] (a superset of ARCH Comp 2023 [67]) on ACC. We began by evaluating safety certification on a small subset of the input space of `ACC_Large` (0.009% of the states verified by $N^3V$) for multiple configurations of each tool (see Table 6). Only NNV was capable of showing safety for 0.1 seconds (vs. time unbounded safety) while taking vastly longer for the tiny fraction of the state space.

**Comparison to NNV.** We performed a more extensive comparison to NNV by attempting to prove with NNV that the NNCS has no trajectories leading from within to outside the loop invariant. This would witness infinite-time safety. Due to a lack of support for nonlinear constraints, we approximate the regions. Over-approximating the invariant as an input region trivially produces unsafe trajectories, thus we can only under-approximate. Notably, this immediately upends any soundness or completeness guarantees (it does not consider all possible NN inputs nor all allowed actions). We apply an

| Tool | Nonlinearities | Evaluated Configurations | Time (s) | Share of State Space | Result |
|---|---|---|---|---|---|
| NNV [93, 94, 97] | no | 4 | 711 | 0.009% | **safe for 0.1s** |
| JuliaReach [18, 85] | no | 4 | — | 0.009% | unknown |
| CORA [7] | yes | 10 | — | 0.009% | unknown |
| POLAR [46] | poly. Zono. | 12 | — | 0.009% | unknown |
| **$N^3V$** | polynomial | 1 | **124** | 100.000% | **safe for $\infty$** |

Table 6: Comparison of verification tools for NNCSs on the ACC Benchmark: Share of state space analyzed and best results of each tool.

interval-based approximation scheme similar to OVERT (detailed in the subsequent paragraph). This scheme is parameterized by $p_{rel}$'s step size ($\sigma$), $v_{rel}$'s distance to the invariant ($\varepsilon$) and the step size for approximating the unsafe set ($\rho$). The right configuration of ($\sigma, \varepsilon, \rho$) is highly influential, but equally unclear. For example, with $\sigma = 0.25, \epsilon = 5$ and $\rho = 1$ we can "verify" not only the retrained `ACC_Large` NN for $2 \le p_{rel} \le 3$, but also the original, unsafe `ACC_Large` *despite concrete counterexamples*. This is a consequence of a coarse approximation, but also a symptom of a larger problem: Neither over- nor under-approximation yields useful results. In particular, discarding inputs close to the invariant's edge equally removes states most prone to unsafe behavior (see Figure 6 in Appendix E.1).

**Approximation Scheme employed for NNV.** We consider input space stripes of width $\sigma > 0$, i.e. $(p_{rel}, v_{rel}) \in [p_0, p_0 + \sigma] \times \left[ -\sqrt{2Ap_{rel}} + \epsilon, \frac{TB}{2} \right]$ ($v_{rel}$ is bounded through the minimally allowed velocity and the maximal velocity that can still decrease $p_{rel}$). While $\sigma$ determines the granularity of the under-approximation of NN inputs, $\epsilon > 0$ discards velocities too close to the loop invariant which cannot be proven using an underapproximation and must be non-zero to prove any system. For each stripe we compute the smallest reachable position $p^*$ and compute a piece-wise linear overapproximation of the negated loop invariant $p_{rel} < v_{rel}^2/(2A)$ on the interval $[p^*, p_0]$ using an approach conceptually similar to OVERT [88]. We determine the number of pieces through a step size $\rho > 0$, i.e. the interval $[p^*, p^* + \rho]$ will have a different line segment than the interval $[p^* + \rho, p^* + 2\rho]$. As the negation of the loop invariant (the orange line in Figure 6) is non-convex, we integrated an iterative check for disjunctions of unsafe sets into the verification procedure of NNV.

**Comparison with DNNV (Table 7)** As DNNV [87] does not support nonlinear properties, a direct comparison to the tool is impossible. However, we improve upon one important feature implemented in DNNV, namely query normalization. By exporting the boolean skeleton generated by MOSAIC, we can use projected model counting [86] to estimate the number of propositionally satisfiable conjunctions over linear constraints. Although the rule based normalization performed by DNNV may produce fewer formulas (this depends on the formula structure and implementation details of the rewriting system), this count provides an upper bound on the number of conjunctions that can be generated for a formula. Without MOSAIC, a rewriting system would first generate a large disjunctive normal form (with at most #Conjunction many elements), then check the feasibility of generated conjunctions and hand feasible conjunctions to an open-loop NNV tool. As indicated by Table 7, such an approach can lead to the number of conjunctions reaching into the trillions which becomes entirely intractable in practice. As can be seen in Table 7, our tool (# Queries) only produces a fraction of the propositionally satisfiable conjunctions (# Conjunctions) and also significantly reduces the number of open-loop NNV queries in comparison to an approach that splits up disjunctions (# Feasible Conjunctions). Note, that DNNV is also required to check generated conjunctions for feasibility, thus, our approach is also efficient in this regard by requiring a comparatively low number of SMT calls. Given the feasiblity of 39 trillion conjunctions, one may wonder whether the propositional structure encoded in the boolean skeleton is of use at all. In this instance, we consider conjunctions over 110 distinct atoms. Indeed, the propositional structure adds value: Without it, we would obtain $2^{110} \approx 10^{33}$ possible conjunctions, i.e. based on the propositional structure we only consider a fraction of approx. $10^{-19}$ of all possible combinations. MOSAIC further reduces this fraction to a degree that is manageable via open-loop NNV.

**Comparison with SMT solvers (Table 8)** An alternative approach for the verification of non-linear open-loop NNV queries could be encoding the problem using an off-the-shelf SMT solver. In this case, the SMT solver has to check the satisfiability of the nonlinear Formula (9). We can instrument

| Property | # Conjunctions | # Queries | # Feasible Conjunctions | # SMT calls |
|---|---|---|---|---|
| ACC | 2.4k | 20 | 86 | 261 |
| ACC (Fallback) | 5.1k | 15 | 72 | 235 |
| ACAS (DNC) | 117.5M | 1.7k | 9.9k | 11.4k |
| ACAS (DND) | 88.9M | 1.8k | 10.4k | 12.0k |
| ACAS (DES1500) | 451.3B | 12.5k | 58.8k | 66.4k |
| ACAS (CLI1500) | 374.4B | 13.1k | 62.5k | 70.4k |
| ACAS (SDES1500) | 9.1T | 18.6k | 64.1k | 75.8k |
| ACAS (SCLI1500) | 18.2T | 21.8k | 76.0k | 88.5k |
| ACAS (SDES2500) | 39.0T | 19.0k | 66.7k | 78.5k |
| ACAS (SCLI2500) | 19.4T | 18.6k | 67.7k | 79.8k |

Table 7: Comparison of feasible conjunctions/queries for non-normalized open-loop NNV queries for an approximation with $N = 1$: #Conjunctions is the number of propositionally satisfiable conjunctions over linear constraints, **# Queries is the number of open-loop NNV queries generated by $N^3V$**, # Feasible Conjunctions is the number of open-loop NNV queries when splitting up disjunctions, # SMT calls is the number of feasibility checks performed by $N^3V$'s MOSAIC implementation during query generation.

| Tool | ACC_Large | | ACC_Large retrained | |
|---|---|---|---|---|
| | Status | Time | Status | Time |
| Mathematica | MO | — | MO | — |
| dReal | TO | — | TO | — |
| Z3 | unknown | 510s | unknown | 1793s |
| Z3++ | unknown | 2550s | unknown | 2269s |
| cvc5 | TO | — | TO | — |
| MathSAT | TO | — | TO | — |
| $N^3V$ | **sat** | **87s** | **unsat** | **124s** |

Table 8: Comparison of $N^3V$ with State-of-the-Art SMT solvers: Timeout (TO) was set to 12 hours

the Lantern package [42] to encode the NN into a SMT formula. Thus, we performed a comparison on the ACC_Large NN as well as the retrained ACC_Large NN, i.e. on a satisfiable as well as a non-satisfiable instance. We compared our approach to dReal [41], Mathematica [96], Z3 [32], MathSAT [27] (due to its use of incremental linearization) as well as the first and second place of SMT-Comp 2023 in the QF_NRA track: Z3++ [24] and cvc5 [14]. The results of our comparison can be observed in Table 8. The observed timeouts after 12 hours are unsurprising insofar as the work on linear open-loop NNV techniques was partially motivated by the observation that classical SMT solvers struggle with the verification of NNs.

**Comparison to the techniques by Genin et al. [42]**    While the work by Genin et al. [42] represents a case-study with techniques specifically applied to an NN for a simplified airborne collision avoidance setting, some ideas from the example in [42] might in principle generalize to other case studies. Unfortunately, the case-study considered by [42] are not the NNs from Julian et al. [53, 55], but simplified NNs with a single acceleration control output. As the authors did not publish their trained NNs, their exact verification formulas, or their verification runtimes, we instead compare our approach with this line of work on our ACC benchmark. To this end, we approximate the verification property derived in Appendix E.1 using the box approximation techniques described by the authors and use their Lantern Python package to translate the verification tasks into linear arithmetic SMT problems. Using Z3, their technique does not terminate within more than 50 hours on the (unsafe) ACC_Large network and thus fails to analyze the NN. This demonstrates significant scalability limitations compared to our approach. Moreover, it is worth pointing out that the authors themselves acknowledge that the technique is incomplete which distinguishes our complete lifting procedure from their approach.

**Results on Q2.**    If closed-loop NNV is a hammer then guaranteeing infinite-time safety is a screw: It is a categorically different problem requiring a different tool. $N^3V$ provides safety guarantees which go infinitely beyond the guarantees achievable with State-of-the-Art techniques (closed-loop

NNV or otherwise). A direct CAD/SMT encoding of Formula (9) as well as the techniques by Genin et al. [42] are no alternatives due to timeouts (>12h) or "unknown" results (see also Table 8).

## E.3 Zeppelin Steering

As a further case study, we considered the task of steering a Zeppelin under uncertainty: The model's goal was to learn avoiding obstacles while flying in a wind field with nondeterministic wind turbulences. This problem has previously been studied with differential hybrid games [77]. The examined scenario serves two purposes: On the one hand, it shows that our approach can reuse safety results from the dL literature which drastically increases its applicability; on the other hand it is a good illustration for why verification (rather than empirical evidence) is so important when deploying NNs in safety-critical fields.

After transferring the differential hybrid games logic contract into a differential dynamic logic contract and proving its safety, we trained a model to avoid obstacles while flying in a wind field with uniformly random turbulences via PPO. After 1.4 million training steps, we obtained an agent that did not crash for an evaluation run of 30,000 time steps. Given these promising results we proceeded to verify the agent's policy assuming a safe – or at least "almost safe" – flight strategy had been learnt. However, upon verifying the NN's behavior for obstacles of circumfence 40, we found that it produced potentially unsafe actions for large parts of the input space. The reason this unsafety was not observable during empirical evaluation was the choice of uniformly random wind turbulences: The unsafe behavior only appears for specific sequences of turbulences which occur extremely rarely in the empirical setting. This flaw in the training methodology was only found due to the verification. This is where our approach differs from simulation-based evaluation: With an SMT filter timeout of 4 seconds, N$^3$V provides an *exhaustive* characterization of *all* potentially unsafe regions in 4.1 hours while providing 72 concrete counterexample regions. This is where our approach differs from simulation-based evaluation, as we were able to generate an exhaustive characterization of counterexample regions. In this instance, the tool's bootlenecks were approximation construction and the SMT based counterexample finding. This case study and the stark difference between simulation and verification underscore the importance of rigorous verification of NNs as an addition to empirical evidence in safety-critical areas.

## F  Allowed advisories for Vertical Airborne Collision Avoidance

Table 9 provides an overview of possible advisories for a Vertical Airborne Collision Avoidance System. The allowed range of vertical velocity and the required minimal acceleration are integrated into the dL model used by VerSAILLE.

| Advisory | Description | Vertical Velocity [ft/min] | Min. Acceleration |
|---|---|---|---|
| COC | Clear of conflict | — | — |
| DNC | Do not climb | $[-\infty, 0]$ | $g/4$ |
| DND | Do not descend | $[0, \infty]$ | $g/4$ |
| DES1500 | Descend at least 1500 ft/min | $[-\infty, -1500]$ | $g/4$ |
| CL1500 | Climb at least 1500 ft/min | $[1500, \infty]$ | $g/4$ |
| SDES1500 | Strengthen descent to at least 1500 ft/min | $[-\infty, -1500]$ | $g/3$ |
| SCL1500 | Strengthen climb to at least 1500 ft/min | $[1500, \infty]$ | $g/3$ |
| SDES2500 | Strengthen descent to at least 2500 ft/min | $[-\infty, -2500]$ | $g/3$ |
| SCL2500 | Strengthen climb to at least 2500 ft/min | $[2500, \infty]$ | $g/3$ |

Table 9: Overview on Vertical Airborne Collision Advisories (simplified version of [50, Table 1])

## G  NMACs produced by NN-based ACAS X advisories

Further counterexamples for the advisories of the NNCS can be found in Figures 7 to 11.

Counterexamples to the safety of NNCS advisories for non-level flight of the intruder in the case of a previous advisory *Do Not Climb* and *Do Not Descend* can be found in Figures 12 and 13. Note,

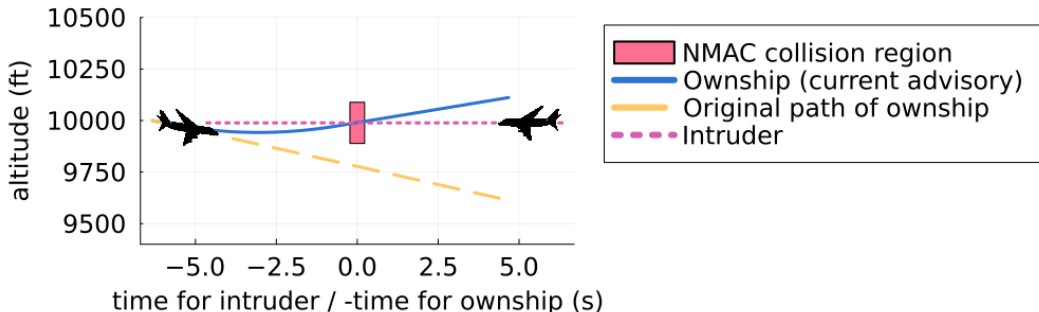

Figure 7: After a previous advisory to descend at least 1500ft/min, the NN advises the pilot to strengthen climb to at least 1500ft/min leading to a NMAC.

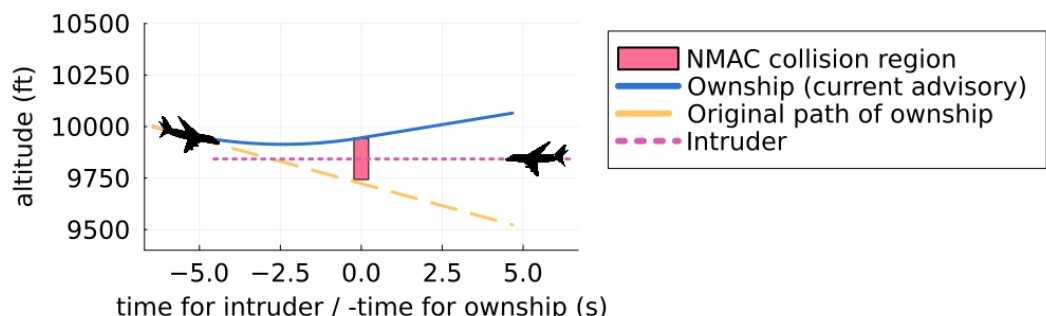

Figure 8: After a previous advisory to strengthen descent to at least 1500ft/min, the NN advises the pilot to strengthen climb to at least 1500ft/min leading to a NMAC.

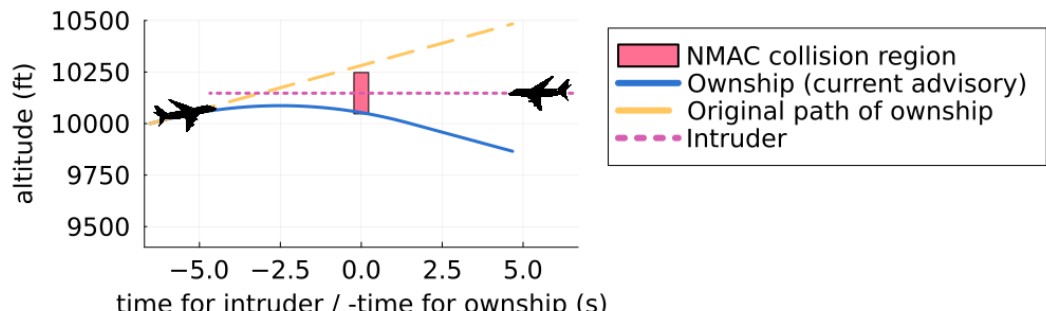

Figure 9: After a previous advisory to strengthen climb to at least 1500ft/min, the NN advises the pilot to strengthen descent to at least 2500ft/min leading to a NMAC.

that for non-level flight (i.e. both intruder and ownship have a non-zero vertical velocity), there exist two possible interpretations for the advised vertical velocities. These can be interpreted as absolute or relative velocity. For our counterexamples in Figures 12 and 13 we opt for the relative velocity interpretation. This does not affect the analysis for level flight intruders where the interpretations coincide.

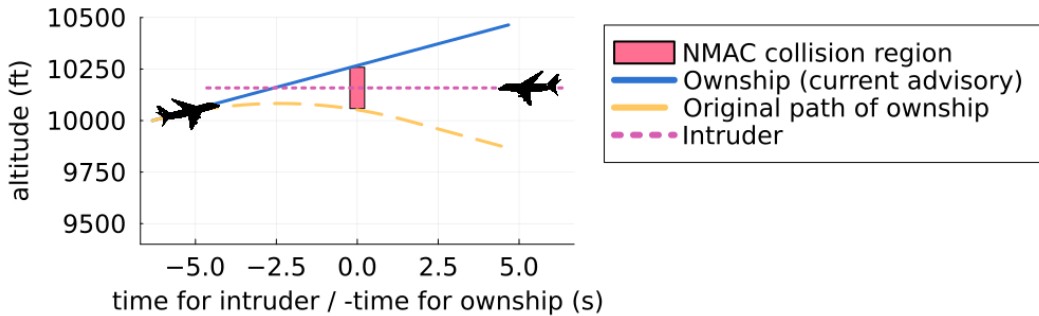

Figure 10: After a previous advisory to strengthen descent to at least 2500ft/min, the NN advises the pilot to strengthen descent to at least 2500ft/min leading to a NMAC.

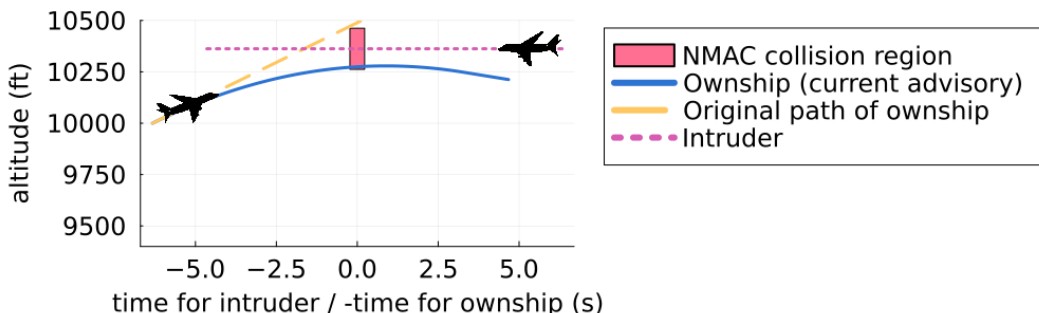

Figure 11: After a previous advisory to strengthen climb to at least 2500ft/min, the NN advises the pilot to strengthen descent to at least 1500ft/min leading to a NMAC.

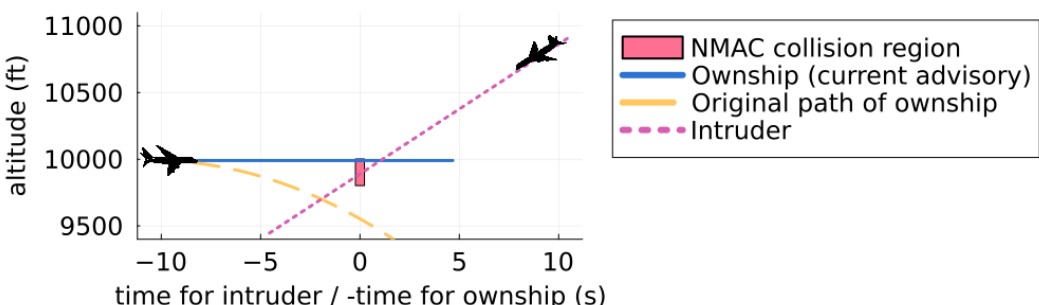

Figure 12: After a previous advisory to not climb, the NN advises the pilot to climb with at least 1500ft/min leading to a NMAC (assumes relative velocity interpretation).

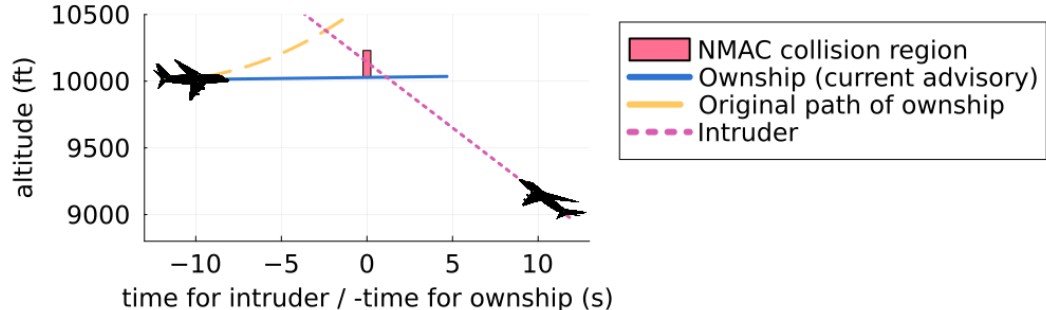

Figure 13: After a previous advisory to not descend, the NN advises the pilot to descend with at least 1500ft/min leading to a NMAC (assumes relative velocity interpretation).

