# OpenReview forum: "Provably Safe Neural Network Controllers via Differential Dynamic Logic"
_NeurIPS.cc/2024/Conference — NeurIPS 2024 poster_

### Official Review · Reviewer_CSRh · 2024-07-09

**Soundness:** 3
**Presentation:** 3
**Contribution:** 3
**Rating:** 7
**Confidence:** 4

**Summary:**

This work addresses the challenge of verifying the safety of NNCSs for CPS, especially for infinite time horizons. To tackle this problem, the author(s) introduce VerSAILLE, a novel approach that leverages differential dynamic logic to derive specifications for NNs, which are then proven using NN verification tools.
This allows proving infinite-time safety of the NNCS via dL.  To bridge the gap between nonlinear arithmetic constraints arising from hybrid systems and linear constraint support in NN verification tools, they also present Mosaic, an efficient, sound, and complete verification method for polynomial real arithmetic properties on piecewise linear NNs. Evaluation shows the effectiveness of VerSAILLE and Mosaic by proving infinite-time safety on benchmarks and enumerating counterexamples in unsafe scenarios.

**Strengths:**

1. The theoretical novelty of this work is sufficient.
  1.1. The proposed VerSAILLE provides sound proof of infinite-time safety for the NNCSs.
       This method reuses safety proofs from control-theory literature in the form of dL for the first time and supports a large class of feed-forward NNs.
  1.2. The framework Mosaic enables the adaptation of existing open-loop NNV tools to handle polynomial queries with arbitrary logical structures while maintaining completeness to polynomial constraints.
2. The experiments are very thorough, especially the comparison and combination with different methods and tools, which proved the effectiveness of the proposed method.
3. This paper is well-written and the running example throughout the article makes it easier to follow.

**Weaknesses:**

My main concern is the gap between the theoretical contributions and the implementation. As mentioned in lines 71-73, "the implementation (N3V) supports NNs with Relu" and "theoretical contribution (VerSAILLE) reaches far beyond this". What are the reasons for this gap and what are the difficulties in overcoming it?

**Questions:**

1. How is the scalability of N3V? What is the structure of the largest neural network it can handle?
2. what is the time overhead of Mosaic i.e., lifting NNV tools for linear, normalized open-loop queries to polynomial queries of arbitrary logical structure?

Please answer the questions in "Weaknesses*" and "Questions*"

**Limitations:**

There are no concerns of negative broader impact.

---

> ### Author Rebuttal · Authors · 2024-08-06
>
> We are glad to see you found our experimental setup to be thorough and consider the paper easy to follow. We are also thankful for your feedback and address your questions and comments below:
>
>
> ## (A) Other architectures & constraints (Weaknesses)
> Like many prior works [40,52,54] we focus our implementation on ReLU NNs.
> This is also the architecture of the NNs by Julian et al. [51,53] -- demonstrating applicability to NNs from the literature.
> Concerning extensibility:
> - For piece-wise linear NNs we can extend our current implementation. Though an engineering challenge, there are no theoretical hurdles. Some piece-wise linear activation functions can be "compiled" to ReLUs as done by DNNV [84] for max-pool layers.
> - For most non-polynomial input/output constraints, we lose completeness due to undecidability. However, Mosaic's framework may help in separating decidable piece-wise linear reasoning from undecidable nonlinear reasoning.
> - For other activation functions (e.g. polynomials or sigmoid, tanh etc.), there exist no complete verifiers. We believe further research will be necessary before our theoretical foundations will manifest in scalable tooling for this setting. Nonetheless, VerSAILLE tells us how to verify such systems once suitable tools become available.
>
> Concerning the latter point, Mosaic fails to generalize to this setting, because we can only simplify the nonlinear arithmetic analysis due to the piece-wise linear behavior of the NN. Without this assumption, the doubly exponential nature of nonlinear arithmetic SMT solving makes the problem very hard to solve without further innovation as we would have to rely on SMT solving (see also comparison in Table 8 Apx. E.2).
>
> ## (B) Scale of NNs (Q1)
> The largest NNs evaluated in our case studies were the NNs by Julian et al. [51,54] which stem from a proposal for a real-world airborne collision avoidance system. The NNs had 6 layers with 45 neurons each, i.e. 270 nodes. This is significantly larger than prior literature on infinite-time safety (e.g. [23, Appendix] consideres single-layer linear NNs analyzed via dReal or [62] considered NNs with approx. 30 neurons; see also comparison to dReal in Appendix E.2). In some NNCS scenarios such as ACAS, medium-size NNs are chosen precisely because they can encode complex control policies while offering a compact representation.
>
> Moreover, we want to emphasize that the verification performed by our approach is not comparable to local verification properties such as, e.g., local robustness: Mosaic has to verify properties w.r.t. the NN's *entire* input space. Naturally, in this context, verifiers scale differently than when "only" verifying epsilon-balls around individual datapoints. In particular, because the verifier must explore larger quantities of feasible ReLU-phase configurations. This difference in scale scale for global properties is also visible in related fields: For example, a very recent State-of-the-Art paper on verifying global robustness properties [R7] is evaluated on NNs with only 50 ReLU neurons.
>
> ## (C) Overhead of Mosaic (Q2)
> Out of the box all evaluated NN verification tools provide a significantly weaker specification language (linear, normalized Open-Loop NNV queries). Consequently, we cannot provide a quantiative measurement for overhead since the properties cannot be verified by the previous tools. If we look at the runtime of the NN verification tool in comparison to the verifier's overall runtime (which then includes all nonlinear reasoning) the share greatly varies by benchmark depending on satisfiability status, query complexity and nonlinearity and can range from as low as 30% to as high as 70%. It is also important to mention that the overhead computation is essential to keeping the NN verification time low: For example, the time spent enumerating disjoint input regions (azulejos) avoids duplicate computations for the NN verification tool (see also our conceptual comparison to DNNV in E.2).
>
> [R7] Athavale, Anagha, et al. "Verifying global two-safety properties in neural networks with confidence." CAV 2024

---

> > ### Comment · Reviewer_CSRh · 2024-08-12
> >
> > Thank you for clarifications, I have no further questions.

---

### Official Review · Reviewer_JF4H · 2024-07-11

**Soundness:** 4
**Presentation:** 4
**Contribution:** 3
**Rating:** 7
**Confidence:** 2

**Summary:**

This paper introduces VerSAILLE, a new method using dL contracts to ensure the safety of Neural Network Controlled Systems with piece-wise Noetherian Neural Networks. Mosaic, implemented in N^{3}V for ReLU NNs, efficiently verifies properties across various case studies, including complex applications like airborne collision avoidance. The approach demonstrates scalability and effectiveness compared to traditional closed-loop techniques, offering a promising path for developing safe and goal-oriented NNCSs in practical settings.

**Strengths:**

The authors have written an good paper with promising contributions which VerSAILLE establishes a formal foundation enabling sound proof of infinite-time safety for NNCSs using dL models from control-theory literature, and  Mosaic introduces an efficient, sound, and complete technique for verifying properties in polynomial real arithmetic on piece-wise linear NNs, enhancing existing open-loop NNV tools. Futhermore, Mosaic supports exhaustive characterization of unsafe state spaces, demonstrated effectively in real-world case studies including adaptive cruise control and airborne collision avoidance ACAS X.

**Weaknesses:**

The structure of the paper is well-organized, with tight logic and thorough arguments, and is overall well-written. However, the introduction and related work sections seem unclear and lack readability. It is recommended to simplify the text or organize it into clearer paragraphs.

**Questions:**

(1) The authors propose a VerSAILLE NNCS verification method, which leverages the efficiency of neural network verification tools while maintaining the rigor of differential dynamic logic. However, the method is only applied experimentally to networks with ReLU activation functions, and most related work focuses solely on this single type of network. Have other types of networks been considered, and has the method been tested for effectiveness across different types NNs?

(2) Additionally, they introduce an effective, sound, and complete method, Mosaic, for verifying polynomial real arithmetic properties on piece-wise linear neural networks, extending existing linear constraint tools to nonlinear settings while preserving completeness. How well does this method work for non-polynomial cases?

(3) Furthermore, in experimental evaluations, the authors demonstrate infinite-time safety in certain scenarios, such as the classical Vertical Airborne Collision Avoidance NNCS verification benchmark.  This benchmark appears frequently in related work; are there additional benchmarks available to showcase, and are there more advanced works for a more comprehensive comparison and validation of the proposed approach?

**Limitations:**

The authors have sufficiently addressed the limitations, explicitly stating the assumptions underlying the theoretical results, particularly in Section 3, and discussing the exponential worst-case runtime of Mosaic in Section 4.

---

> ### Author Rebuttal · Authors · 2024-08-06
>
> Thank you for providing us with the valuable feedback -- we are happy to see you find our approach promising. We address your questions and comments below and are particularly thankful for your feedback on readability:
>
> ## (A) Introduction & Related Work (Weaknesses)
> We address the readability concerns in common answers (A) and (B) and would be happy to hear your feedback on our proposed improvements.
>
>
> ## (B) Other architectures & constraints (Q1, Q2)
> Like many prior works [40,52,54] we focus our implementation on ReLU NNs.
> This is also the architecture of the NNs by Julian et al. [51,53] -- demonstrating applicability to NNs from the literature.
> Concerning extensibility:
> - For piece-wise linear NNs we can extend our current implementation. Though an engineering challenge, there are no theoretical hurdles. Some piece-wise linear activation functions can be "compiled" to ReLUs as done by DNNV [84] for max-pool layers.
> - For most non-polynomial input/output constraints, we lose completeness due to undecidability. However, Mosaic's framework may help in separating decidable piece-wise linear reasoning from undecidable nonlinear reasoning.
> - For other activation functions (e.g. sigmoid, tanh etc.), there exist no complete verifiers. We believe further research will be necessary before our theoretical foundations will manifest in scalable tooling for this setting. Nonetheless, VerSAILLE tells us how to verify such systems once suitable tools become available.
>
> ## (C) Evaluation (Q3)
> We address this concern in common answer (C).
> In particular, we propose to add a summary on our additional experiments to the paper's main section.

---

> > ### Comment · Reviewer_JF4H · 2024-08-10
> >
> > Thank you for your detailed response and promise to clarify the points in the revision. I have no further questions.

---

### Official Review · Reviewer_H7zS · 2024-07-11

**Soundness:** 4
**Presentation:** 4
**Contribution:** 3
**Rating:** 8
**Confidence:** 4

**Summary:**

This paper tackles the challenge of formal verification of neural-network based control systems. While scalability of the existing methods can still be improved, the authors provide an alternative approach via reusing safety proofs from control theory, open-loop neural-network verification, and differential dynamic logic. The authors are mainly concerned with collision avoidance as proxy for safety. Proposed N3V and Mosaic when compared to closed-loop verification tools showed improvement in terms of runtime.

**Strengths:**

The work proposes a novel theoretical framework based on a nontrivial combination of existing theoretical results, previously unemployed by the state of the art. The authors also contribute a technique Mosaic based on a novel algorithmic approach to avoiding repeated checks of the same input regions and instead reusing verification results (azulejos).

The impact of the theoretical framework Versaille extend to the general community of NNCS verification since the core idea of reusing verification proof is generally valuable. The evaluation is extensive enough under the stated assumptions on the activation functions, architecture, input, and safety specification. It would be interesting to see if it extend to other properties and systems. It is a promising step forward to improving verification scalability.

The paper presents technical and theoretical contributions supported by rigorous proofs and extensive evaluation. The evaluation is quite detailed and the authors comprehensively analyze the results in comparison to the state-of-the-art tools.

Overall, the paper proposes a strong theoretical framework for NNCS verification and sound and complete algorithm for the open-loop NN verification. The paper is self-contained and of high quality.



The paper is excellently organized. The authors evidently invested into clear narrative, logical connections, and guidance for the reader.

**Weaknesses:**

One recommendation could be to shortly summarize the take-away from the extended evaluation in the main bod of the paper, so that all evaluation questions are answered without necessarily consulting the appendix. For instance, the Zeppelin insights are quite interesting and probably deserve to to be mentioned in the main paper.

Minor:
- Although "control envelopes" may be a well-understood term in control theory, it would be important to briefly introduce it on the concept level the first time it occurs in the introduction (same applies to differential dynamic logic).
- Illustrative examples and descriptive color names are highly appreciated.
- aircraft
- non-Portuguese speakers would not know how to read "azulejos", would be good to transcribe.
- some abbreviations are not introduced, e.g., DNC (although there original work is cited, it would make the paper self contained to include the definitions).

**Questions:**

1. Mosaic provides an exponential improvement in worst-case runtime. The authors mention that their approach performs particularly well for neural-network control systems with "small" input dimensions. Do the authors have any insights on how small these have to be to perform well, how well is meant here?
2. It seems safety is analyzed here as "collision avoidance" (as it is the case for all considered benchmarks). Do the results generalize to other notions of safety, e.g., stability (can be expressed via reachability as well)?
3. For "unsafe" level flight scenarios, the author mention using manual approximation. To what extend did that contribute to the resulting times in Table 3?
4. Why was the tool compared to ARCH Comp 2022 and not the 2023 results?
5. What is the overhead in preparing system and specifications for N3V and Mosaic? How easily extendable is the implementation?

**Limitations:**

The addressed problem setting is guarded by multiple assumptions, however, common for the NNCS verification domain and evaluated extensively. The check list says the software is open source, however, the authors promise to make it open source, rather than making it open-source anonymously at the time of submission.

---

> ### Author Rebuttal · Authors · 2024-08-06
>
> Thank you for providing us with the valuable feedback -- we are happy to see that you like our proposed approach and found the contents to be well organized.
> We address your questions and comments below:
>
> ## (A) Evaluation (Weaknesses)
> Thank you for the suggestion, we made a proposal for a suitable text in common response (C)
>
> ## (B) Minor Comments (Weaknesses)
> Thank you for the detailed comments which we will integrate in the final version of the paper.
> Concerning your first point, we hope that our updated version of the paragraph from lines 44-53 will help clarify this (see common response (A) ).
>
> ## (C) Complexity of Mosaic (Q1)
> You correctly observed that Mosaic provides an exponential improvement for runtime in comparison to native SMT encoding (p. 7). This is underscored by experimental results (see Table 8 in Apx. E.2) and is what we mean by "well": The problems we solve are beyond the reach of pure SMT.
>
> Since our worst-case complexity is still doubly exponential in the input dimension (but no longer in the NN size), this raises the question how small is "small enough". To this end, there are two answers: To apply the full approach including VerSAILLE, we require an input space which is well-enough understood to be modeled in dL. To apply Mosaic stand-alone, scalability depends on the number of variables and the degree of polynomials. For our case studies, the input spaces had between 2 (ACC) and 4 (Zeppelin/ACAS) dimensions. The output space resp. had dimension 1 (ACC), 2 (Zeppelin) and 9 (ACAS). Due to the unpredictability of SMT solver performance for real-arithmetic we find it difficult to provide reliable guidance in this respect.
>
> ## (D) Supported Properties (Q2)
> The VerSAILLE approach is agnostic to the particular safety property considered:
> We support any property expressible with the box modality ("for all executions ...") in differential dynamic logic.
> Collision freedom is often the most important property in NNCS applications, which is why the examples focus on that.
> Some mistakes in ACAS NN verification in related work stem from settling for partial properties such as
> "If the intruder is distant and is significantly slower than the ownship, [Clear-of-Conflict is among the top choices]" [56].
> While that is a significantly simpler sanity check it does not ensure collision freedom (see also [10]).
>
> ## (E) Manual Approximation (Q3)
> We assume you refer to the following sentence:
> "For unsafe level flight scenarios, we exhaustively characterize unsafe regions going beyond (non-exhaustive) point-wise characterizations of unsafe advisories via manual approximation [54]."
> We did not perform manual approximation in our case study and the times in Table 3 are therefore not impacted by this. Instead, this refers to a characterization of unsafe points (not regions) in a prior work [54] which was based on manual approximation.
> To clarify this, we propose to update the sentence as follows:
> "For unsafe level flight scenarios, we exhaustively characterize unsafe regions. This characterization goes far beyond characterizations in prior work [54] which were generated using manual approximation and resulted in (non-exhaustive) point-wise characterizations."
>
> ## (F) ARCH Comparison (Q4)
> During the initial preparation of the comparison the report on ARCH Comp 2023 was not yet released. Yet the participants of ARCH 2023 are a subset of the participants of ARCH Comp 2022 so we already compare with all participants of ARCH Comp 2023. We will clarify this in the paper's final version.
>
> ## (G) Extensibility of Implementation (Q5)
> Our implementation is modular to support exchanging of different components (e.g. for the use of different NN/SMT solvers). During our implementation, we have already gained some experience with supporting different SMT solvers (though we ultimately decided to stay with Z3 for now) and would be interested to support other tool combinations in the future. We would also be happy to collaborate with interested parties on implementation extensions.
>
> ## (H) Overhead Query Generation (Q5)
> Given a dL model, the generation of the verification property amounts to combining the loop invariant and ModelPlex condition into
> one verification query via an implication. NCubeV supports the syntax produced by ModelPlex (which in turn is the same used for loop invariants).
> Thus, the process is very straight forward.
>
> ## (I) Open-Source Tool (Limitations)
> We provide our implementation with a GPL License in the supplementary materials.
> For the final version, we will link to a GitHub repo containing our implementation with a GPL License. To preserve anonymity we omitted this URL in the draft under review and anonymized the code in the supplementary materials.

---

> > ### Comment · Reviewer_H7zS · 2024-08-11
> >
> > Thank you for clarifications, I don't have further questions. (The proposed paragraphs in the general response contain some tautology.)

---

### Official Review · Reviewer_rNvs · 2024-07-13

**Soundness:** 3
**Presentation:** 2
**Contribution:** 3
**Rating:** 7
**Confidence:** 3

**Summary:**

Authors introduce VerSAILLE (Verifiably Safe AI via Logically Linked Envelopes), a method of verifying neural network based control systems using a provably safe control envelope in dL. VerSAILLE relied on nondeterministic mirrors, which allows authors to reflect a given neural network and reason within and outside dL simultaneously, ultimately having outside dL specifications imply the safety of the mirrored dL model. Authors also introduce Mosaic to leverage off-the-shelf open-loop neural network verifiers on polynomial queries, in addition to their linear ones. Mosaic is both sound and complete, making it a strong fit of critical applications.

**Strengths:**

The paper is comprehensive in its approach and the authors propose a novel method that seems relatively stand-alone. The paper is organized well and the logic of the method flows well.

**Weaknesses:**

Generally, the paper is difficult to read and understand. Readability would be improved from strong definitions and standardized notation.  Paragraphs tend to be very long (notably Related Work), it would be helpful to have more stand alone definitions and methodology.

Page 2, lines 46-53: confusingly written, jumping from point to point without definitions for the terms that are cited. Not sure what the “subsymbolic reasoning of an NN” or “infinite-time control-theory reasoning” are.

Page 2, line 50: “an NN” → “a NN”

Evaluation is conducted only on vertical airborne collision avoidance, and the experiments and parameters are not well defined. On page 8, Table 3, “CE” is not defined.

The use of DPLL(T) is not clearly described in the paper.

POST REBUTTAL:
The authors have cleared up their use of DPLL(T) in their response, which now seems to be an interesting contribution of the work.

**Questions:**

- Page 2, line 56: Is it trivial to assume “abstract, nondeterministic control envelope has already been verified in dL”?
- What are the complexity limitations of Mosaic?

**Limitations:**

Not properly discussed. It would be helpful to understand trade-offs between their methods and other ones.

---

> ### Author Rebuttal · Authors · 2024-08-06
>
> Thank you for providing us with valuable feedback on readability concerns with the paper's draft. We were pleased to hear, that you nonetheless found our approach novel and comprehensive.
> ## (A) Strong Definitions
> > Readability would be improved from strong definitions and standardized notation.
>
> Concerning Introduction and Related Work, see (A) and (B) in our common response.
> We are a little at a loss what you mean by "strong definitions". While we require some new definitions due to the novelty of our work, we do not see this as a deficiency, but as progress and have put significant effort into presenting our approach in a novel unified mathematical framework. Consequently, we provide formal definitions and proofs for all results throughout the paper and appendix while illustrating the concepts via our running example. We strive for standardized notation and appreciate any concrete feedback on aspects that remained unclear. To this end, are there any particular definitions/notations that you would like us to explain and improve upon?
>
> ## (B) Experimental Section
> "CE" stands for counterexamples (will be clarified for Camera-Ready). Similarly we will add explanations for "DNC"="Do Not Climb" etc.
> We will also make use of the additional page for the final version to move appendix details on the experiments into the paper's main section; see common answer (C). The Appendix reports on two additional case studies (adaptive cruise control, zeppelin steering). Additionally, we provide comparisons to other approaches from the literature (e.g. closed-loop techniques and verification via SMT solving). Concerning the well-definedness of experiments and parameters, we would be happy to clarify any open questions. Are there any specific points that you would like clarification on? For the section's final sentence we propose an improved phrasing in our response to Reviewer H7zS; see answer (E) there.
>
> ## (C) Usage of DPLL(T) within Mosaic
> Our original description left all technical details to Apx. B.2.
> We propose to amend llns. 236-251 as follows:
>
> On the technical side, Mosaic proceeds by executing a regular DPLL(T) loop until a satisfiable conjunction of input constraints is found. At this stage, we fix the conjunction's linear constraints (i.e. the azulejo) and an inner loop enumerates conjunctions over mixed/output constraints that are satisfiable in combination with the fixed azulejo. For each such conjunction we save the conjunction of linear mixed/output constraints. This results in a linear, normalized Open-Loop NNV query (conjunction over input, disjunctive normal from over output). We employ a similar inner loop to enumerate satisfiable conjunctions of nonlinear constraints to later check counterexamples via SMT solving (see Retaining completeness). At each step, we interleave propositional and theory solving to discard conjunctions unsatisfiable in real arithmetic as early as possible.
>
> ## (D) dL assumptions (Q1)
> For our approach, dL provides a language to specify desired safety properties together with a model of the system's dynamics and action space. Contrary to other specification languages such as LTL etc., dL directly comes with a technique to *prove* safety w.r.t. the chosen model. Our approach would be equally applicable without a dL proof -- while this yields a less rigorous safety guarantee, it would result in a similar approach to modeling desired requirements in LTL and proving those requirements without a formal safety proof on the abstract level. The proof automation of KeYmaera X [36,86] and recent advancements in control envelope synthesis such as CESAR [55] make it easier to prove correctness for dL control envelopes.
>
> Nonetheless, the triviality of assuming provably safe control envelopes depends on the considered case study: This work aims to make control theory results applicable to NN verification. There are numerous analyses which provide control envelopes for CPS use cases (e.g. [48, R2-R5]). However, our approach cannot escape the undecidability of control system analysis [R1] including NNCS. VerSAILLE makes a large, previously untapped collection of research available to the NNCS verification community, which we believe is a valuable contribution.
>
> ## (E) Trade-Offs between approaches (Limitations)
> We plan to add a summary of our comparative experiments in the paper's main section (see common response). We have made a first attempt at providing insights on the trade-offs between different techniques in this amendment:
> - Closed-Loop tools are very useful for *bounded* time guarantees
> - Our approach can be more efficient by avoiding a reachability analysis for the system's dynamics
> - For prior techniques, approximation-errors make safety proofs impossible close to the controllable region's edge
> - SMT solvers are precise but do not scale to the NNs nor can they alone handle the differential equations
>
> We would be happy about feedback on which further insights would be helpful to understand the literature landscape.
>
> ## (F) Complexity of Mosaic (Q2)
> Mosaic provides an exponential improvement for runtime in comparison to native SMT encoding (see p. 7/ Review H7zS).
> This is underscored by experimental results (see Table 8 in Apx. E.2):
> The problems we solve are beyond the reach of pure SMT. For futher details see also answer (C) for Reviewer H7zS.
>
> [R1] Platzer, André et al. "The image computation problem in hybrid systems model checking." HSCC 2007
> [R2] da Silva, Rafael Rodrigues et al. "Formal design of robot integrated task and motion planning." IEEE CDC 2016
> [R3] Mitsch, Stefan, et al. "Formal verification of obstacle avoidance and navigation of ground robots." IJRR 2017
> [R4] Wu, May, et al. "A Formally Verified Plasma Vertical Position Control Algorithm." FMICS 2020
> [R5] Selvaraj, Yuvaraj et al. "Formal development of safe automated driving using differential dynamic logic." IEEE TIV 2022

---

> > ### Comment · Reviewer_rNvs · 2024-08-12
> >
> > Thank you for your response which answered some of the concerns I had.
> > I have increased my score, though I still have concerns about the limited evaluation of the tool with only 3 case studies.
> > The DPLL(T) description as described is the standard approach that is used, and the description does not indicate the novelty of the proposed solution. If the contribution is an application of existing techniques, then I would expect a much more extensive evaluation (using say all the relevant benchmarks from VNN-COMP).

---

> > > ### Author Response · Authors · 2024-08-12
> > >
> > > Dear Reviewer,
> > > thank you for updating your review based on our answers -- we are happy to hear that we were able to already lift most of your concerns.
> > >
> > > Concerning the usage of DPLL(T) there is a subtle but important difference between classical DPLL(T) and the approach proposed by us:
> > > Classical DPLL(T) takes a formula with arbitrary propositional structure and uses SAT solving on its boolean skeleton to generate *conjunctions* of atoms over a theory which are then handed to a theory solver.
> > > While this approach is extremely useful for classical theory solvers (as they are then not required to analyze complicated propositional formulas), it becomes prohibitively inefficient when used in combination with modern NN verification tools.
> > > The reason for this inefficiency lies in the fact that many NN verifiers perform (somewhat costly) reachablity analysis w.r.t. the NN's input space.
> > > Thus, when a solver supports an interface that allows for a *single* conjunction $C_{\text{in}}$ of constraints in the input space and a *disjunction* of conjunctions $C_{\text{out}}^{1} \lor \dots \lor C_{\text{out}}^{n}$ in the output space,
> > > the verifier can perform the costly reachability analysis w.r.t. $C_{\text{in}}$ *once* and can then use it to simultaneously verify all conjunctions $C_{\text{out}}^{i}$ over the output space.
> > > Therefore, instead of enumerating the conjunctions $\left(C_{\text{in}} \land C_{\text{out}}^{1}\right), \dots, \left(C_{\text{in}} \land C_{\text{out}}^{n}\right)$, our extension of DPLL(T) is capable of generating a *single* query $C_{\text{in}} \land \left(C_{\text{out}}^{1} \lor \dots \lor C_{\text{out}}^{n}\right)$ (see also lns. 238-241) which can make use of this special property of NN verification and, given the cost of NN verification, even an exponential difference.
> > >
> > > The reason we developed this technique lies in the fact that the NNV queries we generate have very complicated propositional structures, e.g. the logical structure of some ACAS specifications have 112 distinct atoms with formula syntax trees of depth up to 9 (see llns.324-325 or also the file `/NCubeV-Artifact-FINAL/test/parsing/examples/acas/property-sdes2500-compressed` of the code submission for a concrete example).
> > > This kind of structure is required for NNCS but otherwise still relatively uncommon in NN verification -- in fact VNN-COMP last year explicitly enforced that all specifications had to be in disjunctive normal form [1, p. 4 "Format"] and we are not aware this rule has been changed in the current iteration.
> > >
> > > This also makes a comparison on VNN-COMP benchmarks less informative, as these benchmarks:
> > > - Only consider linear arithmetic specifications
> > > - Are forced to have an extremely simple propositional structure
> > >
> > > Conversely, our approach was specifically tailored for NNCS benchmarks which:
> > > - contain polynomial arithmetic constraints
> > > - contain very complicated propositional structures
> > >
> > > We hope this can clarify the novelty of our Mosaic procedure in comparison to classical DPLL(T) and thank the reviewer once again for taking the time to review our work.
> > > If there are any further questions we would be happy to follow up.
> > >
> > > [1] https://arxiv.org/pdf/2312.16760v1

---

> > > > ### Comment · Reviewer_rNvs · 2024-08-13
> > > >
> > > > Thank you for the further clarification!
> > > > I have incorporated this into my new score.

---

### Author Rebuttal · Authors · 2024-08-06

We thank the reviewers for their comprehensive reviews and their detailed feedback on the paper. We were excited to hear that the reviewers found our "comprehensive approach" (rNvs) to be "novel" (rNvs,H7zS), "promising" (JF4H) and "well-written" (CSRh). Furthermore we are glad you agree that VerSAILLE is "a promising step forward to improving verification scalability" (H7zS).

Concerning the presentation issues, we plan to augment the presentation with the reviewers' helpful feedback using the additional page granted for the NeurIPS Camera-Ready. To this end, we propose to extend the manuscript as outlined below and look forward to any feedback on our suggestions.

## (A) Readability of Introduction (rNvs,JF4H)
We propose to turn lines 44-53 into an own, updated, paragraph which provides a high-level summary of our approach:

As an alternative to the three outlined approaches, we propose to verify NNCSs based on the rigorous mathematical foundations of differential dynamic logic (dL). dL is a program logic allowing the proof of infinite-time safety for abstract, nondeterministic control strategies (often called *control envelopes*). Due to its expressiveness and its powerful proof calculus, dL even allows the derivation of such guarantees for continuous-time systems or systems whose differential equations have no closed-form solution. By grounding our verification approach in dL, we can reuse safety results from the control theory literature for NN verification -- especially for cases where characterizations of safe behavior and controllable/invariant regions are known (e.g. airborne collision avoidance [48]). How this knowledge can be reused is a non-trivial question: While dL is an excellent basis for reasoning about symbolic control strategies, the numerical/subsymbolic reasoning of NNs at their scale is far beyond the intended purpose of dL's proof calculus. Conversely, open/closed-loop NNV tools and barrier certificates lack the infinite-time or exact reasoning available within dL. This work demonstrates how open-loop NNV can be combined with dL reasoning to combine their strengths while cancelling out their weaknesses. Consequently, by relying on results from the control theory literature, we prove infinite-time safety guarantees for NNCSs that are not provable through either technique alone.

Furthermore, we propose to split the Overview paragraph in line 62 to delineate the contributions of VerSAILLE from the complementary contributions of Mosaic.

## (B) Readability of Related Work (rNvs,JF4H)
We propose to structure the section into the following paragraphs:
- Shielding (lns. 342-346)
- Barrier Certificates (lns. 346-356)
- Open-loop NNV (lns. 356-367)
- Pre-image computation (lns. 367-369)
- Related Techniques (lns. 369-374)
- Closed-loop NNV (lns. 374-379)

If there are further unclear aspects, we would appreciate your feedback.

## (C) Case Studies & Evaluation Section (rNvs,JF4H,H7zS)
Next-generation Airborne Collision Avoidance (ACAS) NNs are standard benchmarks for NN verification, but despite positive verification results [52] counterexamples exist in these NNs even for the vertical case. This underscores that comprehensive closed-loop analysis is crucial for guaranteeing safety. VerSAILLE and Mosaic are first in providing a comprehensive analysis approach for this setting -- despite its complicated nonlinear dynamics. Currently, a major bottleneck for the application of the approach to further case studies is the availability of *safe* neural networks, because sound techniques cannot positively verify unsafe NNCS but merely falsify them. We believe the evaluation in our paper's main part is particularly interesting, because both the NNs [51,53] as well as the dL formalization [48] are from prior literature. That being said, we will add a summary of our results on the ACC and Zeppelin Case Studies from the appendix to the paper's main section. We propose to summarize the additional experiments as follows (moving the Camera-Ready's Appendix to arXiv):

We provide additional experimental results in Appendix E. First, we demonstrate the feasibility of our approach for the running example of Adaptive Cruise Control. Depending on NN size and chosen linearization, our approach can verify or exhaustively enumerate counterexamples for the NNCS in 47 to 300 seconds. For a case study on Zeppelin steering under (uniformly sampled) wind perturbations, we adapt a differential hybrid games formalization [74] to analyze an NN controller trained by us. Here, we encountered a controller which showed very positive empirical performance while being provably unsafe in large parts of the input space: While performing very well on average, the control policy was vulnerable to unlikely wind perturbations -- an issue we only found through our verification. For ACC, we also perform a comparison to other techniques: While Closed-Loop techniques are useful for the analysis of bounded-time safety, their efficiency greatly depends on the system's dynamics and the considered input space. Our infinite-time horizon approach can be more efficient than Closed-Loop techniques as it evades the necessity to analyze the system's dynamics along with the NN (see Table 6). Usually, it is desirable to show infinite-time safety on the entire (controllable) state space. However, the approximation errors incurred via prior closed-loop NNV techniques prohibit this as they will either ignore states inside the controllable region or allow unsafe actions pushing the system outside its controllable region. Conversely, SMT based techniques do not have these approximation issues, but cannot scale to NNs of the size analyzed in this work. We also provide a conceptual comparison demonstrating the efficiency of the Mosaic procedure for normalized query generation over DNNV's expansion based algorithm.

---

### Decision · Program_Chairs · 2024-09-25

**Decision:**

Accept (poster)

**Comment:**

This paper tackles the challenge of verifying safety in Neural Network Control Systems (NNCSs) for Cyber-Physical Systems, particularly over infinite time horizons. It introduces VerSAILLE, a method that leverages differential dynamic logic (dL) to derive specifications for NN controllers, enabling the proof of infinite-time safety. To verify a derived specification against an NN controller, they present Mosaic, an efficient, sound, and complete technique for verifying properties in polynomial real arithmetic on piecewise linear NNs. Mosaic effectively bridges the gap between nonlinear arithmetic constraints from hybrid systems and the linear constraints typically supported by existing NN verification tools. The paper demonstrates the effectiveness of VerSAILLE and Mosaic through several benchmarks and case studies, including adaptive cruise control and airborne collision avoidance.

A limitation of the proposed algorithm is that it produces a safety invariant in dL based on an arbitrary symbolic controller. The safety guarantees are closely tied to the specific characteristics of the symbolic controller. If the NN controller fails to meet the safety requirements associated with this invariant, it suggests that the NN controller may not adhere to the same safety guarantees as the symbolic controller. However, this does not imply that the NN controller is unsafe; rather, it means that the safety assurances applicable to the symbolic controller might not extend to the NN controller. Another limitation of the paper is its exclusive focus on ReLU-based NN controllers. Further exploration in this area could broaden the applicability of the proposed methods.

It is strongly recommended that the presentation, particularly the introduction and related work sections, be improved to better cater to the general NeurIPS audience.